# FIXED POINT EXPLAINABILITY

## ABSTRACT

This paper introduces a formal notion of fixed point explanations, inspired by the "why regress" principle, to assess, through recursive applications, the stability of the interplay between a model and its explainer. Fixed point explanations satisfy properties like minimality, stability, and faithfulness, revealing hidden model behaviours and explanatory weaknesses. We define convergence conditions for several classes of explainers, from feature-based to mechanistic tools like Sparse AutoEncoders, and we report quantitative and qualitative results for several datasets and models, including LLMs such as Llama-3.3-70B.[1]

## 1 INTRODUCTION

The broad adoption of AI solutions in safety critical domains (Jiang et al., 2017; Tambon et al., 2022; Yang et al., 2023) requires the development of explainability techniques that build trust in the end user. In machine learning, explainability and interpretability are concerned with making the decisions and the behaviour of a complex model intelligible (Belle & Papantonis, 2021; Burkart & Huber, 2021) so that humans can interpret, steer, and eventually trust them.

Most of the explainability research in the past decade has focused on neural networks (Samek et al., 2017; Joshi et al., 2021) (although explainability is also relevant for non-neural methods (Izza et al., 2020)), with techniques that range from saliency-based visualisations (Simonyan et al., 2014; Bach et al., 2015; Lundberg & Lee, 2017) to example-based explanations (Li et al., 2018; Chen et al., 2019a; Gautam et al., 2022). Recently, the upsurge in popularity of Large Language Models (LLMs) (Brown et al., 2020) has required the development of techniques to explain their internal behaviour, from Sparse AutoEncoders (SAE) (Bricken et al., 2023; Cunningham et al., 2023) to circuit discovery (Lindsey et al., 2025).

Spanning across more than a decade of research, explainability techniques in machine learning range from low- to high-level (Izza & Marques-Silva, 2024; Zhao et al., 2024), depending on the abstraction of the explanation they return (e.g., a subset of the explained model or a textual explanation), from white- to black-box (Guidotti et al., 2018; Loyola-Gonzalez, 2019) (i.e., whether they explain a model's behaviour by accessing its internal parameters), from causal to correlational (Schölkopf, 2022), etc. In other words, there is no explainer to rule them all, but rather a wide range of techniques and approaches that adapt to different circumstances.

In the process of explaining a complex model, an explainer identifies the reasons behind the decisions of such a model, and so its inconsistencies, errors, and biases (Ignatiev, 2020; Darwiche & Hirth, 2023). As many works outline, explainers are not immune to biases or inconsistencies themselves: recent works show how popular explainers often lack robustness (Alvarez-Melis & Jaakkola, 2018; Dombrowski et al., 2019; La Malfa et al., 2021; Wu et al., 2023; Vadillo et al., 2025), i.e., a small perturbation of the input one wants to explain may result in a very different explanation.

In this work, we question whether an explanation remains consistent across multiple applications of the explainer and what information we can distil from this process. This argument is known in philosophy of science as the "why regress?" principle (Lipton, 2001) and, more broadly, in philosophy, as the "infinite regress argument" (Cameron, 2022). Specifically, the "why regress?" principle posits that recursively *explaining an explanation* makes the causes of a process emerge.

---

[1]The code to replicate the experiments in the paper is available here: `https://anonymous.4open.science/r/fixed_point_explainability_iclr2026-D188`

Figure 1: Left: standard explainability pipeline for an image classifier. Right: how our framework derives a fixed point explanation for the classifier on the left.

Well rooted in human intuition, this principle is a viable instrument to capture the interplay between a model and its explainer.

Building on this intuition, we introduce a formal notion of "why regress?" for explanations by linking it to the mathematical concept of **fixed point**. As Figure 1 illustrates, a fixed point explanation is obtained by recursively applying the explainer to the input and satisfies desirable properties like invariance, minimality, and self-consistency (Alvarez Melis & Jaakkola, 2018). A fixed point explanation can be tested alongside a set of properties, such as the correctness of the model in predicting the right result for each recursive step. If those properties hold upon convergence, a fixed point is a **certificate of stable interplay** of the explainer and the classifier.

In summary, this work introduces the "why regress" as a sanity check for explanations via the notion of fixed point explanation. Our notion can be applied to a several classes of explainers, including feature-, prototype-based, and mechanistic interpretability, for which we define proper convergence conditions. In the experiments, fixed point analysis reveals properties of the model and the explainer that would otherwise remain covered. Specifically, recursive explanations identify, for popular explainability tools and approaches, **inconsistent model's behaviours** and **minimal sets of features** responsible for a model's (mis)classification. We also discuss the theoretical limitations of our framework: in particular, the assumptions one has to make to guarantee convergence to a fixed point for non-monotonic explainers (Paulino-Passos & Toni, 2022), such as rule-based models and LLMs.

## 2 METHODOLOGY

In mathematics, a fixed point is a value that does not change under a given transformation. For an endomorphism $g : X \to X$, a fixed point is defined as $x \in X$ . $g(x) = x$. In explainability, for a generic model $f : X \subseteq \mathbb{R}^a \to Y \subseteq \mathbb{R}^b$, we define an explainer as $\varepsilon : (X; f) \to Z \subseteq \mathbb{R}^c$, or equivalently $\varepsilon : (X, Y; f) \to Z$, as a function that provides some additional information on the model's behaviour. When an explainer is not an endomorphism, one can introduce a *support* function $s : Z \to X$ such that $(s \circ \varepsilon) : X \to X$. Unless strictly necessary, and to ease the notation, we will treat $\varepsilon$ as a deterministic endomorphism: in Appendix A, we provide more details about support functions and their composition with explainers.

**Definition 2.1** (Fixed point explanation). *Given a model $f$ and an explainer $\varepsilon$, a fixed point explanation is defined as $x^* \in X$ for which $\varepsilon(x^*; f) = x^*$ is satisfied.*

A fixed point explanation captures a state of equilibrium between the model and the explainer. This notion is particularly meaningful in interpretability, as it characterizes inputs for which the explanation is faithful, minimal, and stable, as we will show in Section 2.1. Furthermore, motivated by the why regress principle, we introduce the notion of a recursive explanation to formalize the evolution of explanations under repeated application.

**Definition 2.2** (Recursive explanation). *Given a model $f$, an explainer $\varepsilon$ and an input $x$ we are interested in explaining, we define the $k^{th}$-recursive explanation of $f$ w.r.t. $\varepsilon$ as $x_k = \varepsilon(x_{k-1}; f)$, where $x_0 = x$ and $k \geqslant 1$.*

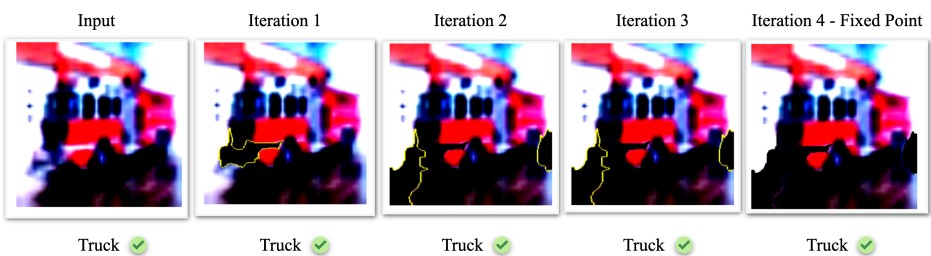

Figure 2: A $P$-fixed point LIME explanation for a VGG16 network (Simonyan & Zisserman, 2015) on CIFAR10 (Krizhevsky et al., 2009). At each iteration, part of the input is occluded (the yellow patches) by the explainer but the model's classification is preserved and correct. After iteration 4 (the fixed point), the correct classification will hold up to infinity.

Intuitively, $x_k$ is the output of the explainer applied $k$ times to the original input $x$. If, at any point in this iteration, the explainer converges to an output that is the same as that of the previous iteration, we observe a fixed point explanation. Namely:

**Definition 2.3** (Fixed point convergence). *Given a model $f$, an explainer $\varepsilon$ and an input $x$ we are interested in explaining, the recursive explanation converges to a fixed point if $\exists k \geqslant 0, \ \forall i \geqslant k$ s.t. $\varepsilon(x_i; f) = x_k$.*

Furthermore, in the process of recursively converging to a fixed point explanation, one can check whether a model satisfies some salient properties $P$ that do not depend on the explanation process, such as the sufficiency of the explanation to imply the correct model's output. Formally:

**Definition 2.4** (A certificate for a fixed point explanations). *For a model $f$, an input $x$, an explainer $\varepsilon$, and a set of properties $P$, $x$ admits a $P$-certificate that holds true **up to infinity** if $\lim_{n \to \infty} \varepsilon(x_n; f) = x_n$ where $\forall k \in \mathbb{N}, \ f(x_k) \models P$.*

In other words, a fixed point explanation that entails a set of properties $P$ will preserve them under any further application of $\varepsilon$, thus acting as a formal certificate for the model. We call this characterisation of Def. 2.1 $P$-**fixed point explanation**. An example of a certificate is *prediction sufficiency* for a classifier, i.e., the features identified by an explainer are sufficient for the model to derive the correct label (La Malfa et al., 2021). An example of an explanation that preserves the model's decision is reported in Figure 2.

## 2.1 On the Benefits of Fixed Point Explanations

This section discusses the properties and benefits of fixed point and $P$-fixed point explanations.

### 2.1.1 The "Why Regress" Argument

The "why regress" argument states that a good explanation can be further questioned with the same arguments to get novel insights (Lipton, 2001). With an explanation that satisfies the "why regress" argument we get to understand more about the model: in this sense, any iteration is *benign*. Fixed point convergence satisfies this principle by construction, as per Def. 2.3. For a $P$-fixed point explanation, the explainer may satisfy or violate, upon convergence, the properties in $P$. The sequence that, upon convergence, satisfies the properties in $P$, constitutes a certificate of such properties **up-to-infinity**, as discussed in the previous section. On the other hand, upon violation of $P$, our framework allows for inspection of the precise iteration where that happened. In Figure 3, we report an example of such analysis and the information one retrieves, upon convergence, if the explanation satisfies or violates the properties in $P$.

### 2.1.2 Optimal Explanations

When an explainer enforces monotonicity, the explanation converges to a minimal explanation w.r.t. any monotone operator.

**Theorem 1.** *For any monotone cost function, fixed point explanations are minimal-cost explanations.*

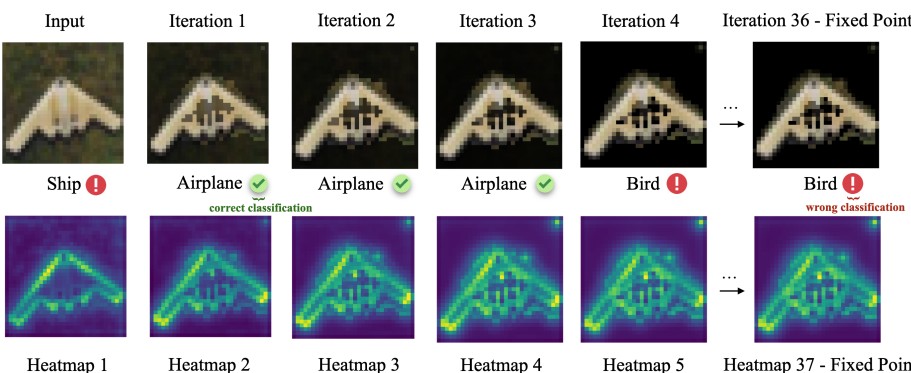

Figure 3: An example of an input that, upon convergence, violates the property of inducing the correct label. The model is a VGG16 network on CIFAR10; the explainer is LRP.

*Proof.* Monotonicity of an explainer on a finite set (of features) and w.r.t. the operator $\subseteq$ is expressed as $A \subseteq B \implies \varepsilon(A) \subseteq \varepsilon(B)$. For any monotone (cost) function $C : X \to \mathbb{N}^+$, $A \subseteq B \implies C(A) \leqslant C(B)$, therefore it holds that $A \subseteq B \implies (\varepsilon(A) \subseteq \varepsilon(B)) \wedge (C(A) \leqslant C(B))$. It follows that $A^* \in \arg\min_{A' \subseteq A} |\varepsilon(A')| \implies A^* \in \arg\min_{A' \subseteq A} C(A')$, i.e., a fixed point for $\varepsilon(A)$ has minimum cost for $C$. $\square$

### 2.1.3 Faithful, Stable, and Progressive Explanations

We argue that $P$-fixed point explanations satisfy the principles of faithfulness and stability as introduced in (Alvarez Melis & Jaakkola, 2018). We also show that one can control the *explanation depth*, i.e., the number of intermediate steps between the input and the fixed point explanation.

**Faithful.** As per (Alvarez Melis & Jaakkola, 2018), faithful means that relevance scores are indicative of "true importance". We assume the "relevance scores" as the features the explainer identifies as important, i.e., $\varepsilon(x; f)$, and as "true importance", a *semantic* condition the model has to satisfy w.r.t. $\varepsilon(x; f)$.

Given an input $x$ and a fixed point explanation for $\varepsilon(x; f)$, let's denote as $\Delta x_i = (x_{i+1} \setminus x_i)$ the left-out features at the $i^{\text{th}}$-iteration of $\varepsilon$. We also denote the iteration where the explanation converges as $k \in \mathbb{N} . x_i = x_{i+1}, \forall i \geqslant k$. Now, as a measure of "true importance", let's define a set of properties $P$. It is then true that for a fixed point explanation $\forall i \in [1, k], f(x_i \setminus \Delta x_i) \models P$. In other words, the left-out features at each iteration are irrelevant to imply $P$ and, by exclusion, what remains upon convergence is indicative of "true importance", up to the limit of the explainer, as in Figure 2.

**Stable.** We interpret the notion of stability in (Alvarez Melis & Jaakkola, 2018) as the invariance of the explanation and/or the model, to local perturbations of the input. This notion has been extensively studied in other works (Alvarez-Melis & Jaakkola, 2018; Dombrowski et al., 2019; Ghorbani et al., 2019a; La Malfa et al., 2021; Vadillo et al., 2025). For a $P$-fixed point explanation, that can be easily expressed as a property of $P$. For example, the invariance of the model to local perturbations (Madry et al., 2018), e.g., in $\mathcal{L}_p$-norm, can be expressed as $f(x') = y, \forall x' . ||x - x'||_p \leqslant \rho$, and enclosed in the set of properties $P$. A similar argument applies to the stability of the explainer. We nonetheless notice that such properties are computationally hard to verify (Katz et al., 2017).

**Explanation depth.** Our fixed point and $P$-fixed point explanations are defined to return, at each iteration, an input that (i) the model can further process, and, for $P$-fixed point explanation, (ii) is property preserving. One can further control the *explanation depth*, i.e., how many intermediate steps to return upon convergence, to distil as much information as possible from the process. Under the assumption that the explainer is continuous w.r.t. its parameters $\delta$, one can extend Defs. 2.2- 2.4 to make the progression continuous, i.e., $x_{k+\delta} = \lim_{\delta \to 0} \varepsilon(x_k, \delta; f)$. In Figure 4, we show an example of a SHAP explanation, where we increase the depth and discover that the explainer, while refining the explanation, induces the classifier to predict a different class.

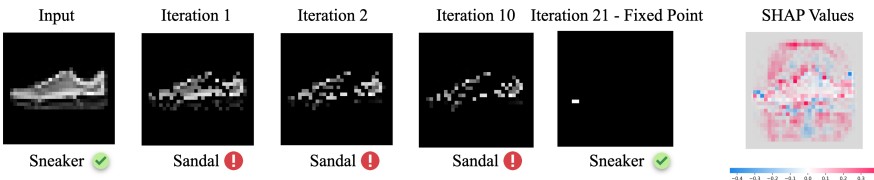

| Input | Iteration 1 | Iteration 2 | Iteration 10 | Iteration 21 - Fixed Point | SHAP Values |
|---|---|---|---|---|---|
| Sneaker ✓ | Sandal ❗ | Sandal ❗ | Sandal ❗ | Sneaker ✓ | |

Figure 4: An example of how a fixed point SHAP explanation can enhance a FashionMNIST (Xiao et al., 2017) explanation by augmenting the explanation length (for a VGG16 net). While the input is correctly classified at the beginning, the successive refinements violate induce the wrong label for the classifier. The process eventually converges to a very succinct explanation that is nonetheless classified correctly.

## 3 CLASSES OF FIXED POINT EXPLAINERS

This section shows on what assumptions one can guarantee the existence of a fixed point explanation for different classes of classifiers.

### 3.1 FEATURE-BASED EXPLAINERS

For a generic model $f : X \to Y$, a feature-based explainer returns a subset of the input features as an explanation, i.e., $\varepsilon : X \to \{1, 2, \cdots, |X|\} \subseteq I$. A support function allows for processing the explanation recursively, namely $s : (X, I) \to X$. A common choice for a support function preserves the values of the features in the explanation and zeroes the left-out features, e.g., $s(x, I) = (x^{(i)}$ if $i \in I$, 0 otherwise).

**Theorem 2.** *Any finite, $\subseteq$-monotonic, feature-based explainer admits, for any input, a fixed point explanation.*

*Proof.* Let $(I, \subseteq)$ be a complete lattice and $(\varepsilon \circ s) : I \to I$ a monotonic function w.r.t. $\subseteq$. Then the set of fixed points of $(\varepsilon \circ s)$ in $I$ forms a complete lattice under $\subseteq$. (Tarski, 1955)  □

In other words, for any input, under monotonicity of $(\varepsilon \circ s)$ w.r.t. the $\subseteq$ operator, i.e., $A \subseteq B \implies \varepsilon(s(A)) \subseteq \varepsilon(s(B))$, a fixed point always exists (note that even non-monotonic explainers will behave monotonically when combined with e.g. masking-based support functions, ensuring convergence). We conjecture similar guarantees may hold even without the monotonicity assumption, which we will pursue as a future direction.

In terms of desirable properties a $P$-fixed point explanation can satisfy, one is the aforementioned *sufficiency of the input*, as Fig. 2 illustrates, as well as robustness of the model and the explainer to local perturbations. In the experiments, we test fixed point and $P$-fixed point explanations for widely employed explainability tools, including LIME (Ribeiro et al., 2016), SHAP (Lundberg & Lee, 2017), and LRP (Bach et al., 2015).

### 3.2 PROTOTYPE-BASED EXPLAINERS

Prototype-based explanations justify a model's prediction by selecting inputs from a finite set of prototypes $S$, grounding the decision in how the input resembles interpretable and representative examples. These explanations typically consist of a subset $\varepsilon(x) \subseteq S$, such as the $k$ nearest prototypes or those satisfying a criterion (e.g., class agreement). To operate in a semantically meaningful space, inputs $x \in X$ are embedded into a latent space $\mathcal{Z}$ using an encoder $e : X \to \mathcal{Z}$. This enables to optimize the prototypes directly in the latent space, which are then visualized through a decoder $d : \mathcal{Z} \to X$, and to define similarity through a distance functions $\delta : \mathcal{Z} \times \mathcal{Z} \to \mathbb{R}$.

For recursion, we rely on a support function $(s \circ \varepsilon)(x) = \arg\min_{p \in \varepsilon(x)} \delta(x, p)$ that returns the prototype closest to the input being explained. Since $(s \circ \varepsilon)(x) \in S$, it trivially follows that only prototypes in $S$ can be fixed points. However, being a fixed point also requires that the prototype remains its closest prototype after decoding and re-encoding, a property we term *self-consistency*:

(a) ProtoVAE (Gautam et al., 2022)

(b) PrototypeDNN (Li et al., 2018)

Figure 5: Self-consistency patterns for two prototype-based models. Each image correspond to one decoded prototype, and arrows indicate the nearest prototype according to the model's similarity function.

**Definition 3.1** (Self-Consistency). *A prototype-based explainer satisfies self-consistency for $p \in S$ if $\arg\min_{p' \in S} \delta([e \circ d](p), p') = p$.*

**Corollary 2.1.** *A prototype $p \in S$ is a fixed point only if self-consistency is satisfied for $p$.*

We experimentally check this relevant property in Section 4.2, finding that even state-of-the-art prototype-based models fail to satisfy it (see Fig. 5).

**Convergence of recursive explanations** The "why regress" principle states that an explanation should itself be explainable using the same mechanism. In this context, we show that recursive prototype-based explainers induce a sequence of prototypes which always converge to either a (i) fixed point, or, more generally, (ii) a cycle of length $m$:

**Theorem 3.** *Given a finite prototype set $S$, a deterministic model $f$ and a deterministic explainer $\varepsilon$, then $\exists n, m \in \mathbb{N}^+$, s.t. $\forall k \geqslant n$, $\varepsilon(x_k; f) = \varepsilon(x_{k+m}; f)$, where $x_k$ is the prototype at iteration $k$.*

Theorem 3 also holds when self-referential explanations are prevented by excluding the previous prototype at each step. The proofs for both scenarios are shown in Appendix B.1. We note that determinism is a soft assumption, satisfied in practice by most existing prototype-based models (Chen et al., 2019a; Gautam et al., 2022; Hase et al., 2019; Li et al., 2018). Further theoretical results are provided in Appendix B.2, including conditions for class-preserving cycles. Experimental results validating self-consistency violations and class-preserving cycles are provided in Section 4.2 and Appendix B.3.

### 3.3 MECHANISTIC INTERPRETABILITY AND SPARSE AUTOENCODERS

The notion of fixed point explanation can also be applied to mechanistic interpretability tools; in particular, we focus on LLMs and SAE. A SAE is an AutoEncoder trained to extract interpretable and "steerable" features from dense hidden representations of an LLM. The SAE training paradigm forces its hidden representations to be sparse, so that only few interpretable features are activated at each forward step.

As sketched in Figure 6, to interpret a generic LLM that predicts a token out of a finite vocabulary $V$, i.e., $f : V \to \mathbb{P}(V)$, a SAE $\varepsilon$ takes as input an $f$ hidden activation and returns a reconstruction of the hidden representation $z \in Z$, namely $\varepsilon : (Z, f) \to Z$. The SAE learns to explain the LLM by mapping a model's hidden activation into a high-dimensional, *interpretable* vector $h \in H$. Of particular interest for the experimental part are the $P$-fixed point explanations, in relation to the tokens the LLM would predict with the iterated explanation, namely:

**Definition 3.2** (SAE $P$-fixed point explanation). *For an LLM $f : V \to \mathbb{P}(V)$, a SAE $\varepsilon : (Z, f) \to Z$ has a $P$-fixed point explanation for an input $x$ when $\lim_{n \to \infty} \varepsilon(z_n; f) = z_n$ where $\forall k \in \mathbb{N}$, $f(x| do(z \leftarrow z_k)) \approx f(x| do(z \leftarrow z_{n+1}))$.[2]*

That is, a $P$-fixed point explanation preserves the next-token prediction of the LLM: the condition $\mathbb{P}(f(x| do(z \leftarrow z_k))) \approx \mathbb{P}(f(x| do(z \leftarrow z_{n+1})))$ guarantees that the token distribution at each

---

[2]The "*do*" refers to the classic intervention in causal inference, also known as "patching".

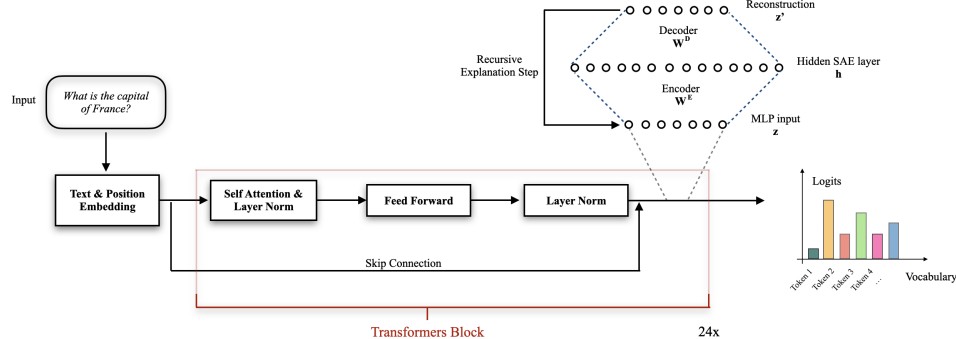

Figure 6: A prototypical LLM's Transformers block and the prediction for a given input. A fixed point explanation is obtained by recursively applying the SAE feature extraction (top), after the first Transformers layer norm, until convergence.

iteration changes minimally. While a lossless SAE has, trivially, for any input, a fixed point that converges after one iteration, in Appendix C we expand on the general conditions to guarantee the existence and uniqueness of a fixed point and $P$-fixed point explanation, for both the linear and the non-linear case. In the experiments, we show how popular pre-trained SAEs lack consistency as the features' reconstruction does not preserve the LLM predicted tokens, posing serious issues regarding trusting their explanations and the "disentangled" features they are supposed to extract.

## 4 EXPERIMENTAL EVALUATION

### 4.1 FEATURE-BASED EXPLAINERS

On three standard explainers in literature, namely LIME, SHAP, and LRP, and three standard datasets, namely MNIST, FashionMNIST, and CIFAR10, we compute fixed point and $P$-fixed point explanations on a VGG16 network. Quantitative results for each dataset and explainer are reported in Table 1. LIME explanations are very short and inconsistent (i.e., the recursive steps do not preserve the class predicted by the model originally): we impute this to LIME being a local explainer. On the other hand, LRP and SHAP explanations are more consistent, as they act globally and are less affected by spurious features in the input. Figure 7 reports $P$-fixed points for each class in a dataset, revealing a minimal subset of the anchoring features of each class. Further results for different support functions are provided in Appendix A.

### 4.2 PROTOTYPE-BASED EXPLAINERS

As discussed in Section 3.2, a prototype $p$ is guaranteed to be a fixed point as long as the model is *self-consistent* (see Definition 3.1). We evaluated this condition for two popular prototype-based neural architectures introduced in (Gautam et al., 2022) and (Li et al., 2018). Results reveal that this property does not consistently hold in practice, as we show in Figure 5: for both models, $\exists p \in S$ such that $\arg\min_{p' \in S} \delta\big((e \circ d)(p), p'\big) \neq p$. This reveals that, despite being a crucial property for reliable self-explainable models, state-of-the-art models do not inherently satisfy self-consistency. We also observed that self-consistency degrades significantly as the number of prototypes increases, which introduces a trade-off between predictive performance and consistency (see Table 4 in Appendix B.3). These findings suggest an opportunity for future research to investigate whether enforcing self-consistency through explicit regularisation during training can improve model robustness, reliability, and interpretability without degrading the predictive performance.

### 4.3 SAE

We focus on Llama-3.3-70b and Llama-3.1-8B (Grattafiori et al., 2024), which we explain with a open-source SAEs (McGrath, 2024; EleutherAI, 2024). We compute fixed point and $P$-fixed point explanations (Def. 3.2) on a dataset containing three types of questions: numerical, multiple choice,

| Dataset | Explainer | # Features | # Features (fixed point) | Self-Consistency | Convergence Steps |
|---------|-----------|-----------|--------------------------|------------------|-------------------|
| MNIST | LIME | 1,024 | 20.48 ± 61.44 | 15% | 3.29 ± 0.45 |
| | SHAP | 1,024 | 133.12 ± 75.77 | 30% | 9.74 ± 2.39 |
| | LRP | 1,024 | 1013.76 ± 10.24 | 97% | 2.25 ± 0.54 |
| FashionMNIST | LIME | 1,024 | 3.48 ± 33.8 | 12% | 3.14 ± 0.35 |
| | SHAP | 1,024 | 71.68 ± 40.96 | 22% | 14.87 ± 3.85 |
| | LRP | 1,024 | 686.08 ± 174.08 | 67% | 3.99 ± 0.61 |
| CIFAR10 | LIME | 150,528 | 45,158 ± 30,105 | 36% | 7.02 ± 2.31 |
| | SHAP | 150,528 | 19,065 ± 473 | 52% | 10.0 ± 0.0 |
| | LRP | 150,528 | 46,663 ± 6,021 | 26% | 5.34 ± 0.92 |

Table 1: Fixed point explanation metrics for LIME, SHAP, and LRP. The model explained is a VGG16 trained on each dataset. More details about the model's performance in Appendix E.1.

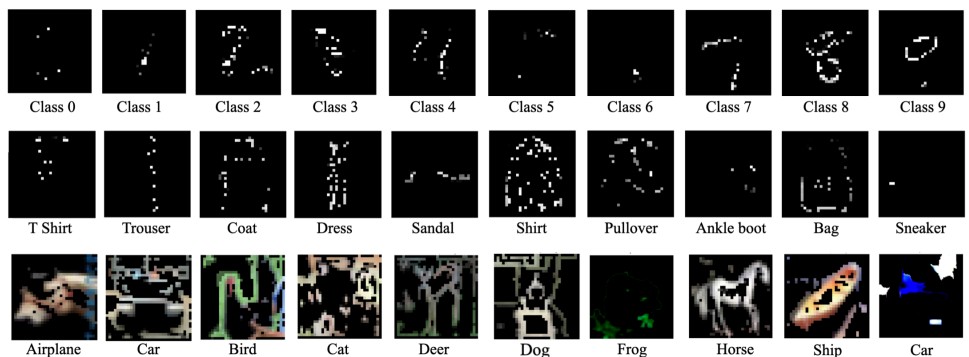

Figure 7: Fixed point explanations of MNIST (top), FashionMNIST (centre) and CIFAR10 (bottom) classes, that preserve the correct model's classification upon convergence and up to infinity.

and general knowledge. The datasets has been synthetically generated by GPT4 (OpenAI, 2023) to cover domains requiring a single-token answer. More details are provided in Appendix C.1. We apply the SAEs in the deeper layers of LLMs, where the skip connections of transformer layers accumulate features across the model. Table 2 shows that the SAEs converge to a fixed point 100% of the time, and do so faster and towards more informative fixed points across larger models, hinting at a scaling law for stability and recursive explanation quality. In particular, we see that consecutive SAE iterations of Llama 70b have lower initial Jaccard similarities, converging faster to the FPE (defined as total overlap), and lower KL divergence with the initial logits, which yield 93% P-FPE in numerical questions. In Appendix C.2, we further report results for Llama-3.1-8B.

## 5 RELATED WORK

**Explainable AI and interpretable explanations.** Explainability techniques in machine learning can be grouped into several categories depending on the interplay between the model and the explainer, the type of explanation returned, how *intrusive* the explainer is into the model, etc. (Molnar, 2020; Murdoch et al., 2019).

*Post hoc* methods (Retzlaff et al., 2024) return explanations after training and usually requires an external model; popular techniques include LIME and Anchors (Ribeiro et al., 2016; 2018). Conversely, *ante hoc* methods concern intrinsically explainable models such as decision trees or self-explainable neural networks (Alvarez Melis & Jaakkola, 2018; Gautam et al., 2022; Izza et al., 2020; Li et al., 2018; Narodytska et al., 2018).

A *black-box* explainer depends solely on the input and the output of the model, with provided explanations that are usually sufficient to imply the decision of the model, as for the case of Rule-Fit (Friedman & Popescu, 2008); conversely, a *white-box* explainer accounts for the model and the process that produces the output: that is the case for gradient-based explainers such as LRP (Bach et al., 2015) and most of the mechanistic interpretability methods such as SAE and circuit discovery techniques (Bricken et al., 2023; Cunningham et al., 2023; Lindsey et al., 2025). Other explainers

| Question Type | Setting | # Iterations | Correctness | Correctness (Top5) | Jaccard | KL Div |
|---|---|---|---|---|---|---|
| **Numerical** | base | 0 | 0.6436 | 0.9703 | - | - |
| | single | 1 | 0.4158 | 0.7525 | - | 0.6665 |
| | double | 2 | 0.3069 | 0.5941 | 0.3264 | 1.0862 |
| | constant | $7 \pm 3$ | 0.0000 | 0.0000 | 0.0020 | 5.6397 |
| **Multiple-Choice** | base | 0 | 0.9700 | 0.9900 | - | - |
| | single | 1 | 0.9700 | 0.9900 | - | 0.1482 |
| | double | 2 | 0.9700 | 0.9900 | 0.2786 | 0.2242 |
| | constant | $7 \pm 1$ | 0.9300 | 0.9900 | 0.0022 | 0.6239 |
| **General-Knowledge** | base | 0 | 0.2647 | 0.4706 | - | - |
| | single | 1 | 0.0686 | 0.2059 | - | 0.9323 |
| | double | 2 | 0.0098 | 0.1275 | 0.3413 | 1.7299 |
| | constant | $13 \pm 12$ | 0.0196 | 0.0980 | 0.0019 | 5.3277 |

Table 2: Metrics for FPEs of Llama-3.3-70B produced by an SAE across three categories of questions. Setting specifies how many SAE iterations are applied to the LLM, i.e., 0, 1, 2, or the #Iterations sufficient to get to the constant point. **Correctness** (the proportion of correct answers for the top token, or the top five tokens) **is our $P$-fixed point explanation in the constant case**. Jaccard similarity is the proportion of overlapping features with the first SAE iteration. The KL divergence is between original logits and those patching the SAE reconstruction in the residual stream.

combine causal inference with explainability (Madumal et al., 2020; Beckers, 2022), to identify the necessary part of an input that guarantees the model to behave in that way (Watson et al., 2021), or, complementarily, the minimal perturbations of the input that induce an error (Goyal et al., 2019; Guidotti, 2024).

Closer to our work, (Ghorbani et al., 2019b; Gorokhovatskyi & Peredrii, 2021; Fel et al., 2023) incorporate recursive procedures as part of their explanation methods, to decompose concepts or iteratively subdivide visual inputs to enhance interpretability. In contrast, our framework applies recursion to the explanations to assess their stability, convergence, and self-consistency. Complementarily, a series of works provide theoretical and practical tools to assess the faithfulness of neural network explainers (Camburu et al., 2019; Jacovi & Goldberg, 2020; Slack et al., 2021).

**Guaranteed explainability.** Guaranteed explainability is concerned with formally explaining the decisions of a complex model. In the context of classic machine learning algorithms, methods focus on exact explanations (Marques-Silva et al., 2020; Izza & Marques-Silva, 2021), while for neural networks researchers are more interested in explanations that are, for the model, robust to adversarial manipulations (Paul et al., 2024; Wu et al., 2024). Standard techniques for neural network models encompass abduction based methods (Ignatiev et al., 2019; La Malfa et al., 2021; Bassan et al., 2025) and counterfactual explanations (Jiang et al., 2022; Leofante & Lomuscio, 2023).

The aforementioned works mainly focus on robust explanations, i.e., those guaranteed to remain invariant for the model. A complementary line of research devises robust explainers, i.e., methods that provide explanations that are robust for the explainer (Chen et al., 2019b; Dombrowski et al., 2022; Liu et al., 2022; Wang et al., 2020; Wicker et al., 2022). A few works analyse common explainers and how their output changes when explaining different models and initial conditions (Apley & Zhu, 2019; Kantz et al., 2024).

## 6 CONCLUSIONS AND FUTURE WORK

We introduce the notion of fixed point explanation as the result of the recursive interaction between a model and its explainer. $P$-fixed point explanations refine this notion by acting as certificates for explanations with respect to a set of properties that will hold up to infinity. Fixed point explanations satisfy minimality and faithfulness, and provide theoretical characterisations for several classes of explainers, including feature-based explainers and SAEs. Our framework also identifies failures, inconsistencies, and interesting interplays between a model and its explainer that would not otherwise emerge. These findings advance our understanding of the limitations of current explainability techniques and open the door to new evaluation frameworks grounded in stability and recursion, motivating fixed point explanations as a valuable lens for future research on robust explainability.

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

APPENDIX

## A  EXPLAINERS, ENDOMORPHISMS, AND SUPPORT FUNCTIONS

When an explainer is not an endomorphism, one needs a support function $s$ that, in composition with the explainer, makes the composition $\varepsilon \circ s$ an endomorphism so that the notion of fixed point explanation applies. For prototype-based explanations, as well as SAEs, the support function is not required as the explainability process is an endomorphism. As we discuss in the main paper and in Appendix D.1 and D.2, rule-based explainers and LLMs require a careful characterisation of the support function, as characterising them as monotonic explainers misses the core characteristics of the models we would like to explain. We thus leave their characterisation, as well as a theoretical extension of our framework, as future work.

In contrast, for feature-based explainers, designing good support functions is inherently challenging: a well-designed support must effectively suppress discarded features without excessively distorting the input or reintroducing discarded features (which would disrupt the recursive filtering process). To better understand the sensitivity of our framework to this choice, we conducted additional experiments exploring alternative support functions for feature-based explanations on CIFAR-10. More detailedly, we compared the standard zero-out strategy against three alternatives designed to balance feature suppression and visual coherence:

- **Exponential Decay**: masked pixel values are multiplied by a decay factor (0.7), progressively fading discarded regions.
- **Class-wise Average Filling**: masked pixels are replaced by class-specific average images, providing a more semantically plausible background.
- **Gaussian Blur**: masked pixels are replaced by a locally blurred version of the original image, which preserves low-frequency structure while removing fine details.

We focus the analysis on the LRP explainer, due to its computational efficiency and higher flexibility in the explanation masking. Quantitative results are shown in Table 3. As it can be observed, similar consistency levels are obtained across all methods (differences within approximately 2%-12%) while being less disruptive visually (e.g., Gaussian blurring), demonstrating that our results are robust to the choice of the support function. Figure 8 provides qualitative comparisons, illustrating how each support method influences the appearance of recursive inputs throughout the process.

## B  PROTOTYPE-BASED EXPLAINERS

### B.1  PROOF FOR THEOREM 3

Theorem 3 states that, given a finite prototype set $S$, a deterministic model $f$ and a deterministic explainer $\varepsilon$, then $\exists n, m \in \mathbb{N}^+$, s.t. $\forall k \geqslant n$, $\varepsilon(x_k; f) = \varepsilon(x_{k+m}; f)$,, where $x_k$ is the prototype at iteration $k$.

*Proof.* The proof is based on the observation that the process of selecting $x_{k+1}$ at each step $k$ is deterministic, assuming determinism for $f$ and $\varepsilon$. These transitions can be deterministically characterized by the current prototype, $x_k \in S$, by a transition function $T : S \to S$, precisely, $T(x_k) = \big(s \circ \varepsilon(x_k; f)\big)$, based on the operators defined in Section 3.2. This also applies to the

| Metric | Zero-out | Exp. Decay | Class-wise Averaging | Gaussian Blur |
|---|---|---|---|---|
| **Self-consistency** | 26% | 26% | 14% | 24% |
| **Convergence Steps** | $5.34 \pm 0.92$ | $69.48 \pm 3.42$ | $6.92 \pm 1.20$ | $6.64 \pm 1.34$ |

Table 3: Self-consistency rates (%) and convergence steps of LRP on CIFAR-10, under different support functions: zero-out masking, exponential decay (with a factor of 0.7), class-wise average filling and Gaussian blur.

scenario in which, at each step $k$, the current prototype is excluded (thereby preventing self-referential explanations). Since, in both cases, $T$ is **deterministic**, and the number of possible states is **finite**, there must exist a period $m$ bounded by $|S|$ in which the path traversed becomes cyclical. This can be proven by the fact that $T$ induces a functional digraph, in which every node has an outdegree of 1, which implies that **(i) every connected component has one directed cycle and (ii) all edges that are not in a cycle are pointing towards the cycle** (Alon et al., 2010). □

An alternative proof can be given based on the pigeonhole principle. Since the prototype set $S$ is finite, and the explainer selects each iteration a prototype from $S$, the sequence must eventually revisit a previously selected prototype. Once this occurs, the deterministic nature of the process ensures that the same sequence of prototypes will repeat, resulting in a cycle.

### B.2 CLASS PRESERVATION

It can be shown that recursive prototype-based explanations satisfy the *class preservation* property under some mild conditions. Let us assume that the set of prototypes can be partitioned into $b$ subsets $S_1, \ldots, S_b$, where $b$ is the number of classes of the problem and $p \in S_i$ implies that $p$ belongs to the $i$-th class. Let us also assume that the input $x_0$ belongs to the class $s$, yielding a closest-prototype $x_1 \in S_s$. A sufficient condition for the recursive process (when self-reference is allowed) to converge to a class-preserving sequence of prototypes is:

$$\forall p \in S_s, \ \min_{p' \in S_s} \delta(e \circ d(p), p') \leqslant \min_{q' \in S \setminus S_s} \delta(e \circ d(p), q').$$

Similarly, this condition can be straightforwardly reformulated for the case in which self-reference is prevented, by replacing $S_s$ by $S_s \setminus p'$ in the left hand side of the inequation. We experimentally validate class preservation in Appendix B.3 for different prototype-based explainers and configurations.

### B.3 EXPERIMENTAL RESULTS FOR PROTOTYPE-BASED EXPLAINERS

To evaluate our theoretical framework, we conduct an empirical study using two influential prototype-based architectures: ProtoVAE (Gautam et al., 2022) and PrototypeDNN (Li et al., 2018) (hereinafter *ProtoDNN*, for simplicity). These models are particularly well-suited to our analysis, as they define class-level explanations in terms of entire prototypes, rather than relying on fine-grained, local features that explain only part of the input (Chen et al., 2019a; Hase et al., 2019; Carmichael et al., 2024). We trained both architectures on the MNIST and Fashion-MNIST datasets, and for an increasing number of prototypes: $|S| \in \{10, 20, 50, 100\}$. The models have been trained using the original implementation released by the authors. Our experiments focus on three key aspects, summarized in Table B.3:

- *Self-consistency (Self-Cons.)*: Whether each prototype is closest to itself after decoding and re-encoding, reporting the percentage of prototypes that satisfy this property.

- *Class preservation*: Whether recursive explanations preserve class membership. We evaluated this property for recursive explanations starting from (i) prototypes themselves or (ii)

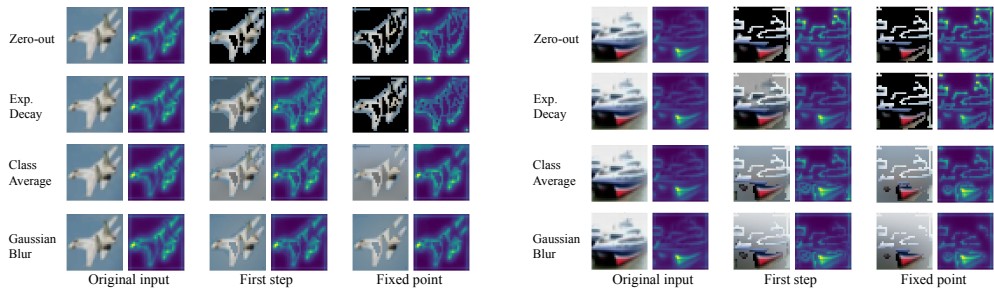

Figure 8: Visual comparison of different support functions for CIFAR10 and LRP. Each row shows a different support method: zero-out, exponential decay, class-wise averaging, and Gaussian blur.

| Model | Dataset | $|S|$ | Acc. | Self-Cons. | Class-P. $(S, \circlearrowleft)$ | Class-P. $(S, \cancel{\circlearrowleft})$ | Class-P. $(X, \circlearrowleft)$ | Class-P. $(X, \cancel{\circlearrowleft})$ | Steps $(\circlearrowleft)$ | Steps $(\cancel{\circlearrowleft})$ |
|---|---|---|---|---|---|---|---|---|---|---|
| ProtoVAE | MNIST | 10 | 94.1% | 100% | 100% | 0% | 99.8% | 0.0 | 1/1.0/1 | 2/2.2/3 |
| ProtoVAE | MNIST | 20 | 99.1% | 60% | 100% | 100% | 99.9% | 99.9% | 1/1.3/2 | 2/2.0/2 |
| ProtoVAE | MNIST | 50 | 99.2% | 32% | 100% | 100% | 99.9% | 99.9% | 1/1.5/4 | 2/2.7/5 |
| ProtoVAE | MNIST | 100 | 99.2% | 18% | 100% | 100% | 99.9% | 99.9% | 1/1.6/3 | 2/2.7/5 |
| ProtoVAE | F-MNIST | 10 | 66.2% | 100% | 100% | 50% | 84.9% | 26.0% | 1/1.0/1 | 2/3.9/6 |
| ProtoVAE | F-MNIST | 20 | 90.9% | 35% | 90% | 90% | 72.4% | 72.4% | 1/1.9/3 | 2/2.3/3 |
| ProtoVAE | F-MNIST | 50 | 91.6% | 22% | 100% | 96% | 82.8% | 79.0% | 1/1.9/4 | 2/2.8/6 |
| ProtoVAE | F-MNIST | 100 | 91.6% | 8% | 99% | 99% | 79.8% | 79.8% | 1/2.0/3 | 2/2.9/4 |
| ProtoDNN | MNIST | 10 | 98.8% | 100% | 100% | 40% | 79.5% | 21.2% | 1/1.0/1 | 2/2.4/3 |
| ProtoDNN | MNIST | 20 | 98.9% | 90% | 100% | 80% | 99.4% | 59.5% | 1/1.1/2 | 2/2.6/4 |
| ProtoDNN | MNIST | 50 | 99.0% | 84% | 100% | 94% | 99.0% | 85.8% | 1/1.1/2 | 2/2.8/5 |
| ProtoDNN | MNIST | 100 | 98.5% | 75% | 100% | 100% | 99.2% | 99.2% | 1/1.0/2 | 2/2.9/5 |
| ProtoDNN | F-MNIST | 10 | 90.1% | 80% | 100% | 30% | 70.5% | 9.4% | 1/1.2/2 | 2/2.4/3 |
| ProtoDNN | F-MNIST | 20 | 89.5% | 85% | 90% | 60% | 68.2% | 39.0% | 1/1.1/2 | 2/2.8/4 |
| ProtoDNN | F-MNIST | 50 | 89.7% | 70% | 86% | 54% | 78.1% | 48.5% | 1/1.1/3 | 2/3.8/7 |
| ProtoDNN | F-MNIST | 100 | 89.6% | 67% | 91% | 73% | 80.4% | 62.2% | 1/1.2/5 | 2/3.8/7 |

Table 4: Experimental validation of prototype-based recursive explanation properties, evaluated for different models, datasets and number of prototypes $|S|$. **Acc.** represents the test accuracy. **Self-Cons.** reports the percentage of prototypes that are their closest neighbour. **Class-P.** reports class preservation for recursive paths starting at each prototype ($S$) or from a test input ($X$), evaluated allowing ($\circlearrowleft$) and preventing ($\cancel{\circlearrowleft}$) self-references in two consecutive steps. **Steps** reports the *min./mean/max.* number of steps before convergence to a cycle for recursive explanations starting from test inputs.

    test inputs. In both cases, we report the percentage of cases in which recursion preserves the class predicted at the starting point. We compute this percentage for the entire set of prototypes in case (i), and for the test set in case (ii).

- *Convergence Steps*: Number of recursion steps before convergence to a cycle when explaining test inputs. For completeness, we report the minimum, average and maximum number of steps (computed for the set of test inputs).

In order to measure the effect of self-references, we evaluated the last two metrics both when allowing ($\circlearrowleft$) and preventing ($\cancel{\circlearrowleft}$) prototypes from pointing back to themselves in the recursion. Furthermore, metrics involving input samples have been evaluated for the entire test set of the corresponding dataset, averaged taking into account only the inputs correctly classified by the model.

As can be seen in Table B.3 self-consistency decreases significantly as the number of prototypes increases. This also introduces a trade-off between predictive performance and consistency, which is particularly significant for ProtoVAE. These results point to a promising direction for future work: exploring whether introducing explicit regularization to enforce self-consistency during training can enhance the robustness, reliability, and interpretability of the model—without compromising its predictive accuracy.

Despite varying levels of self-consistency, prototypes preserve class membership during recursion in most configurations when the recursion begins from prototypes. At the same time, class preservation degrades substantially when self-references are disallowed across consecutive steps, especially when the number of prototypes is small, as this naturally limits the ability to sustain a *why-regress* recursion. This is also reflected in the number of recursion steps required to converge, which generally increases with the number of prototypes. Finally, we note that, when recursion starts from test inputs, class preservation drops further. This can be attributed to the fact that classification relies on the entire distribution of prototype distances, so the closest prototype selected for explanation may not belong to the predicted class, particularly in the case of ambiguous or borderline inputs. An example of such a prediction is shown in Figure 9.

Figure 9: Prototype-based explanation for an ambiguous input (leftmost), followed by its 10 nearest prototypes in the latent space, ordered by similarity (distance shown above each prototype). Although the model (ProtoVAE, $|S| = 50$) classifies the input as a 5, the three closest prototypes resemble a different class, highlighting a case where nearest prototypes may not align with the predicted label.

## C  SAE

### C.1  ON CONTRACTIVE SAE AND FIXED POINT EXPLANATIONS

In this section we expand on the general conditions to have a fixed point and a $P$-fixed point explanation, as discussed in Def. 3.2.

We first address the simplified linear case, and we then discuss the non-linear case. As sketched in Figure 6, we assume the SAE is a two-layers encoder-decoder that computes $\varepsilon(z) = a(zW^E)W^D$, where $z$ is the input to the SAE as computed by $f$, $W^E$ and $W^D$ are the weight matrices associated to the encoder and decoder, and $a$ is an activation function: a common choice for $a$ is the binary top-$k$, which selected the $k$ features with largest absolute activation.

**The linear case.**    A linear SAE has the following mathematical formulation: $\varepsilon(z; f) = zW^EW^D$, which we refer to as $zW$ for the linearity of the transformations involved. A condition to have a fixed point in a recursive SAE is that the SAE transformation $W = W^EW^D$ is "contractive". In other words, that is equivalent to checking whether (i) $W$ is diagonalisable and (ii) its spectral norm is lower or strictly lower than 1.[3]

Suppose all the eigenvalues of $W$ are at most 1. In that case, the repetition of the application of the SAE will converge to zero for all the dimensions that correspond to the eigenvectors with eigenvalues $< 1$. In contrast, they will converge to a non-zero fixed point for those equal to 1.

Final note. If there is one single eigenvalue with value 1, and (i) and (ii) hold, then the fixed point is unique and the same for any input; otherwise, it is different for each input (that's what we expect in general by a non-trivial linear SAE). If an eigenvalue is complex, we look at its norm: if it's complex but its norm is $< 1$, the process goes to zero (in a spiral and via rotations). If it equals 1, it rotates forever and does not converge. If it is $> 1$, it diverges.

In Figure 10, we show the convergence (divergence) of AutoEncoders that by construction are contracting (expansive).

**The non-linear case.**    A non-linear SAE has the following mathematical formulation: $\varepsilon(z; f) = a(zW^E)W^D$, with $a$ the binary top-$k$ function. Proving that an input has a fixed point requires testing that any combination of the possible $\binom{|h|}{k}$ binary vectors the SAE produces has a fixed point, i.e., that the decoder is contractive. While that requires a combinatorial number of steps, there are a few alternative approaches to mitigate the hardness of the problem. To guarantee the global convergence of any input, one can estimate the Lipschitz constant of the SAE; the $inf$ and $sup$ of each SAE input dimension $z^{(i)}$ give a bound of the Lipschitz constant of the network, from which one can derive the max and min of each output neuron analytically. While a global Lipschitz bound is hard to estimate, or too loose in practice, one can assume the Hamming distance between a hidden representations $h$ and its recursive iteration $h' = \text{top-}k(hW^DW^E)$ being bounded (e.g., only a few neurons of the top-$k$ may change between the two).

---

[3]The spectral norm of a diagonalisable projection $W = A\Sigma A^T$ is the square root of its maximum eigenvalue. In the general case of a non-squared matrix, one would look at the Lipschitz of the transformations (easy for the linear case and bounded by the norm of the matrix), or at other techniques such as SVD.

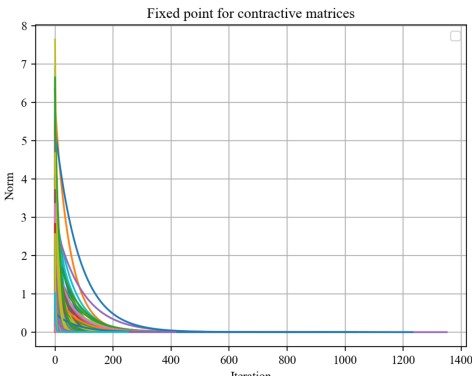 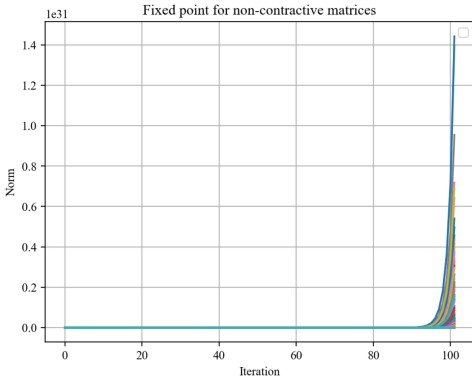

Figure 10: Convergence/divergence of the recursive process of a linear SAE $\varepsilon(x) = xW$. Left: 1000 random 10x10 matrices $W$ that, by construction, have their eigenvalues with norm at most 1. Right: 1000 random 10x10 matrices that, by construction, have some of their eigenvalues with norm larger 1.

| Question Type | Setting | # Iterations | Correctness | Correctness (Top5) | Jaccard | KL Div |
|---|---|---|---|---|---|---|
| **Numerical** | base | 0 | 0.5446 | 0.9802 | - | - |
| | single | 1 | 0.6238 | 0.9505 | - | 0.1517 |
| | double | 2 | 0.6139 | 0.9505 | 0.8720 | 0.1910 |
| | constant | $29 \pm 12$ | 0.0000 | 0.0000 | 0.0169 | 10.0825 |
| **Multiple-Choice** | base | 0 | 0.7300 | 0.9700 | - | - |
| | single | 1 | 0.7600 | 0.9700 | - | 0.1071 |
| | double | 2 | 0.7500 | 0.9900 | 0.8653 | 0.1680 |
| | constant | $17 \pm 4$ | 0.0000 | 0.0000 | 0.0268 | 9.7767 |
| **General-Knowledge** | base | 0 | 0.1667 | 0.3529 | - | - |
| | single | 1 | 0.1863 | 0.3627 | - | 0.0819 |
| | double | 2 | 0.1667 | 0.3529 | 0.8831 | 0.1145 |
| | constant | $31 \pm 13$ | 0.0000 | 0.0000 | 0.0182 | 8.7088 |

Table 5: MMetrics for FPEs of Llama-3.1-8B produced by an SAE across three categories of questions. Setting specifies how many SAE iterations are applied to the LLM, i.e., 0, 1, 2, or the #Iterations sufficient to get to the constant point. **Correctness** (the proportion of correct answers for the top token, or the top five tokens) **is our $P$-fixed point explanation in the constant case**. Jaccard similarity is the proportion of overlapping features with the first SAE iteration. The KL divergence is between original logits and those patching the SAE reconstruction in the residual stream.

### C.2 ADDITIONAL EXPERIMENTS WITH SAE

In Table 5, we report the results for the fixed point explanations we obtained with Llama-3.1-8B. For reference, results for Llama-3.3-70B are reported in the main paper, Table 2.

## D NON-MONOTONIC EXPLAINERS

In this section, we discuss the notions of fixed point and $P$-fixed point explanation for two classes of explainers that, we believe, should not be subjected to the constraint of monotonicity in the recursive step. While we believe their characterisation constitutes a valuable research direction, this goes beyond the scope of this work and we thus leave it as future work.

### D.1 RULE-BASED EXPLAINERS

Let's consider a model $f : X \to Y$, $|Y| < \infty$. An explainer is defined as $\varepsilon : X \to R$, $|R| < \infty$. Suppose $X \in \{0, 1\}^d$ and $R = \{x^{(i)} = \{0, 1, ?\}\}$, where $x^{(i)}$ is the i-th feature of $x$ and $\{0, 1, ?\}$ means that its value is either assigned to 0 or 1 or undefined. For example, a valid assignment for a

bi-dimensional input is $\{0, ?\}$, which corresponds to the rule 'if $x^{(1)} = 1$ then $y$'. Given a support function $s$ (e.g., all the points undefined in $R$ are replaced by the mode of that variable in the training data), to guarantee convergence one has to prove, or construct $\varepsilon$ and $s$ such that $(\varepsilon \circ s)(x)$ is, for any perturbation $s$ can induce for undefined variables in an assignment $r$, a **stable attractor**.

We now provide a definition of fixed point explanation for a rule-based explainer:

**Definition D.1** (Non-monotonic fixed point explanation for rule-based explainers). *For a generic model $f : X \rightarrow Y$, an explainer that returns a finite set of rules $r \in R$, i.e., $R = \{x^{(i)} = \{0, 1, ?\}\}$, and a support function $s : R, X \rightarrow X$ fixed point explanation is defined as $\lim_{n \to \infty} (s \circ \varepsilon)(x; f) = r_n$.*

Since the choice of the support function heavily influences the characterisation of a fixed point, and we drop the assumption regarding the monotonicity of the explainer, we leave the characterisation of a fixed point for rule-based systems as future work.

## D.2 LARGE LANGUAGE MODELS

LLMs are powerful models used as general-purpose assistants in many tasks, from code completion to complex reasoning (Brown et al., 2020). With the recent upsurge of reasoning models (DeepSeek-AI, 2025; Kumar et al., 2024), LLMs are trained to imitate humans in their interactions, which are known to be non-monotonic (Antoniou & Williams, 1997): humans and LLMs in fact "jump" from an incomplete set of observations to a conclusion and may retract them in the light of new observations (Paulino-Passos & Toni, 2022).

To adapt the notion of fixed point and $P$-fixed point explanation, one needs to carefully consider these observations and design an explainer so that (i) it provides an explainability process that converges to a stable state while (ii) not denaturalising the way they produce rationales, i.e., in free-form text.

**Definition D.2** (Non-monotonic fixed point explanation for LLMs). *For an LLM that outputs a sequence of tokens until a termination condition is met, i.e., $f : x \in V^+ \rightarrow y \in V^+$,[4] an explainer that acts on the concatenation of the LLM input-output and returns a textual explanation of such sequence, i.e., $\varepsilon : (x \oplus y) \in V^+ \rightarrow r \in V^+$, a fixed point explanation is defined as $\lim_{n \to \infty} (\varepsilon \circ f)(x \oplus y) = r_n$.*

Differently from the other definitions of fixed point and $P$-fixed point explanations, here we employ both the model and the explainer; of the latter, we discard the monotonicity assumption, as we argue it is an ill-posed condition for reasoning models (Paulino-Passos & Toni, 2022). We argue for $(f \circ \varepsilon)$ being a **stable attractor** that converges, for good explainers, to a fixed point. That implies that an explanation converges to a fixed point after eventually inflating the explanation.

We leave the theoretical characterisation of a fixed point for inflated LLMs explanations as future work.

## E EXPERIMENTAL DETAILS

In this section, we report further experimental details of our work. In terms of computational resources, the experiments on feature-based and prototype-based explanations have been conducted on low-end computers (e.g., laptops without GPUs and a limited amount of RAM, e.g., $16GB$). As regards the experiments on SAEs, we have been granted access to high-end computational resources by a company (in particular, one H100), and we thank that company in the acknowledgments.

### E.1 FEATURE-BASED EXPLAINERS

**Models and training details.** For each dataset, a VGG16 model is trained for 5 epochs, with an accuracy of $0.94$ on MNIST, $0.85$ on FashionMNIST, and $0.88$ on CIFAR10. MNIST and FashionMNIST images are scaled from 28x28 to 32x32 pixels to be compatible with the VGG16 network by padding their border.

---

[4]To be precise, an LLM computes this function $f : V^+ \rightarrow \mathbb{P}(V)$, and predicts a sequence in an auto-regressive fashion.

## E.2 SAE

**Representative question examples.**

- **Multiple-choice**
    1. Which planet is closest to the sun? A=Mercury B=Venus C=Earth D=Mars Answer=
    2. What is the capital of Germany? A=Hamburg B=Berlin C=Munich D=Frankfurt Answer=
    3. How many legs does a spider have? A=6 B=8 C=10 D=12 Answer=
- **Numeric**
    1. $7^2 =$
    2. the square root of 16=
    3. How many legs does a dog have=
- **General Knowledge (one token answer)**
    1. What is the capital of France?
    2. What do bees produce?
    3. What is the opposite of cold?

