# OpenReview forum: "Fixed Point Explainability"
_ICLR.cc/2026/Conference — Submitted to ICLR 2026_

### Official Review · Reviewer_WWxa · 2025-10-29

**Soundness:** 3
**Presentation:** 3
**Contribution:** 2
**Rating:** 6
**Confidence:** 2

**Summary:**

This paper introduces fixed point explanations; iteratively explaining explanations from a model for a given input-label pairing until convergence. This follows the idea of humans asking 'why?' to a provided explanation, attempting to develop and understanding of the answer provided. This can reveal the models inconsistent behaviours, captured in the variability between explanations, and minimal sets of features responsible for a models behaviour.

**Strengths:**

- The notion of recursive explanations is interesting and of interest to the ICLR community.
- The authors justify why the principles of recursive explanations are of importance in explainable AI (XAI).
- The authors provide reasoning as why to why the recursive explanations satisfy the commonly used properties of faithfulness and stability, although the applications differ from the traditional methods in XAI.
- The experiments cover a range of modalities.

**Weaknesses:**

- Section 2.1.3 could benefit from added discussion on the key differences between the principles applied in traditional XAI, and in the fixed point iteration. For example, stability in an algorithm such as LIME is usually seen as perturbations in the feature space or perturbations in the sampled neighborhood when training your local model, and how does this related with the fixed point iteration definition?
- I also think it would help readers familiar with XAI to see some simple visual examples - for example a 2-d example iterating using a feature-based explainer. It would help relate to concepts that people in the field are familiar with.

**Questions:**

- As in the weaknesses section, what do the authors think are the main differences between aspects of explainers such as stability and faithfulness in the traditional XAI sense vs the fixed point iteration setting?
- Do these aspects behave similarly to the traditional XAI defining, following areas of uncertainty in the models classification? Different behaviour can be observed depending on whether the data point you are explaining is close to the decision boundary or far. It would also be interesting to see the movement. I could also see some of these aspects being captured by a sphere which surrounds the explanations in all iterations in feature space for a feature-based explainer, i.e. if the radius of this sphere is large then there is more variability between iterations.

---

> ### Author Response · Authors · 2025-11-22
>
> We thank the reviewer for the time they spent on articulating their concerns. We kindly ask them, before reading the rebuttal, which addresses all the points other reviewers raised, to read the message to all the reviewers and the area chairs.
>
> **W1. Section 2.1.3 could benefit from added discussion on the key differences between the principles applied in traditional XAI, and in the fixed point iteration.**
> *For example, stability in an algorithm such as LIME is usually seen as perturbations in the feature space or perturbations in the sampled neighborhood when training your local model, and how does this related with the fixed point iteration definition?*
>
> - This is a point on which we agree. LIME has been extensively studied under the lens of perturbations and local robustness. In our framework, this notion is considered if one applies a P-fixed point explanation and explicitly enforces robustness. We discuss that at the end of Section 3.1. On the other hand, local robustness is hard to compute (it’s known it is np-hard for relu networks), and that would make the computation of the fixed points slower.
>
> - On the other hand, for standard fixed point explanations, the notion of local robustness is not considered, as what matters for convergence is what is the attractive point that results in the interplay between a model and its explanation (that point can be a singleton if the fixed point is unstable).
>
> **W2. I also think it would help readers familiar with XAI to see some simple visual examples - for example a 2-d example iterating using a feature-based explainer. It would help relate to concepts that people in the field are familiar with.**
>
> - We will add more explicative figures of the convergence process in the revised version of the article.
>
> ------
>
> **[QUESTIONS]**
>
> **Q1. As in the weaknesses section, what do the authors think are the main differences between aspects of explainers such as stability and faithfulness in the traditional XAI sense vs the fixed point iteration setting?**
>
> - As we discuss in the point above (W1), fixed point interactions do not necessarily guarantee a solution to be robust to perturbations. That can be enforced with a P-fixed point. In terms of faithfulness, introducing the requirement that the model should always predict the same class for each iteration relates to the notion of faithfulness in XAI. All these aspects are described in the paper, but we are happy to improve them.
>
> **Q2. Do these aspects behave similarly to the traditional XAI defining, following areas of uncertainty in the models classification?** *Different behaviour can be observed depending on whether the data point you are explaining is close to the decision boundary or far. It would also be interesting to see the movement. I could also see some of these aspects being captured by a sphere which surrounds the explanations in all iterations in feature space for a feature-based explainer, i.e. if the radius of this sphere is large then there is more variability between iterations.*
>
> - This aspect is indeed interesting, and we plan to run some experiments where we enforce local robustness of each fixed point, until convergence. If at some point an explanation is not robust, that would break one requirement of the P-fixed point so defined, and can be studied and compared to the previous iteration to understand which features made the model brittle (in this sense, that could be related to the notion of minimal adversarial attack).

---

### Official Review · Reviewer_uFZV · 2025-10-31

**Soundness:** 2
**Presentation:** 4
**Contribution:** 3
**Rating:** 4
**Confidence:** 5

**Summary:**

This paper introduces fixed point explainability, a theoretical framework that formalizes the stability of explainability methods through recursive application. The authors argue that an explainer ε, when repeatedly applied to its own outputs with respect to a model f, should converge to a fixed point x* satisfying ε(x*; f) = x*. Such convergence is interpreted as a certificate of self-consistency between the model and its explanation. To generalize this idea, the paper defines P-fixed points, where the recursion preserves a set of properties P that maintains model's predictive invariance. Theoretical guarantees for the existence of fixed points are provided under determinism and monotonicity assumptions.

The framework is instantiated for three families of explainers. First, feature-based methods are examined to test whether their attribution maps remain stable under recursion. Second, prototype-based models are analyzed, showing that prototype interactions often lead to cyclic rather than convergent behavior. Third, the concept is extended to SAEs used in mechanistic interpretability of large language models, where recursive interventions are used to test consistency of token-level predictions. Across these domains, the experiments show that many explainers fail to reach consistent fixed points, implying that current explanation techniques may not be self-consistent with the models they are meant to interpret. The authors propose that fixed-point analysis can thus serve as a general diagnostic tool for evaluating the stability and internal validity of explanation methods.

**Strengths:**

The paper presents a conceptually novel and intellectually stimulating approach to evaluating explainability methods. Its central idea, that recursion of an explainer on its own outputs can reveal internal consistency between the model and the explanation, introduces a fresh perspective to a field. The mathematical framing is clear and well motivated. The authors carefully define fixed points and property-preserving fixed points, connect them to the fixed-point theorem, and articulate how convergence may serve as an internal validity check. The unifying scope of the framework is also impressive. It brings together three interpretability directions, e.g, feature-based attribution methods, prototype-based reasoning models, and sparse autoencoder analysis in LLMs, under a single analytical lens. This breadth demonstrates that the idea is not bound to a specific architecture but reflects a general principle about model–explainer interactions. The paper is written with strong clarity and coherence, and the theoretical–philosophical motivation adds depth without overshadowing the formalism. Overall, it provides a thought-provoking conceptual foundation that could inspire a new class of diagnostics for stability and faithfulness in explainability research.

**Weaknesses:**

**1.**
The core premise equates recursive stability with explanatory validity. This is conceptually fragile. A trivial constant explainer that outputs the same mask or rationale for every input reaches a fixed point in one step, yet it communicates no model specific information. Stability, as defined here, is therefore not sufficient for faithfulness. The paper acknowledges faithfulness informally, but never rules out such degenerate fixed points nor provides a practical test that separates meaningful convergence from vacuous convergence.


**2.**
The theoretical guarantees rest on determinism and monotonicity. Many widely used explainers are stochastic or adaptive. LIME samples neighborhoods and can change across runs. SHAP often relies on sampling estimates. Under such settings, non convergence is expected behavior due to intrinsic randomness, not a failure of explanation. One could study convergence in expectation or concentration around a mean fixed point, but this is not developed. As a result the negative empirical findings risk re describing stochasticity rather than revealing lack of faithfulness.

**3.**
The recursion requires that the explainer output can be reinjected as a valid input. This domain alignment rarely holds in practice. For saliency or GradCAM the output is a heatmap, not an input image. The paper introduces support functions to coerce reinjection, yet these choices change the induced dynamics and can dominate the result. In the appendix the authors try several masking strategies and obtain different behaviors. This sensitivity implies that the fixed point is as much a property of the chosen support as of the explainer or the model. The main text does not quantify this dependence.

**4.**
Recursive application can collapse information and force convergence for the wrong reason. If each step removes or attenuates features, the process can converge to empty or near empty representations. Such convergence is a signature of information decay, not of explanatory adequacy. The paper shows shrinkage for some feature explainers, but treats this as evidence of instability rather than as a predictable artifact of the masking operator. Without a conservation or sufficiency control, convergence is ambiguous.

**5.**
The prototype analysis treats cycles as a pathology, yet cycles can reflect meaningful conceptual neighborhoods. A pair or small set of prototypes can be mutually nearest under the model similarity metric while still representing coherent local structure. Declaring non convergence as failure ignores this possibility. The theory anticipates cycles in a finite state system, but the empirical section does not attempt to measure whether cycles align with human concepts or with class structure.

**6.**
For LLM SAEs the property P is defined as preservation of token level distributions under residual patching. The experiments show large intermediate drift before eventual stabilization. This demonstrates that recursion can degrade predictions in the process, which limits its utility as a diagnostic that one would actually run. Moreover, token preservation is a model internal criterion. There is no check of semantic coherence of features or alignment with human judged rationales. The claim that P fixed points certify anything beyond internal invariance is therefore unsubstantiated.

**7.**
External validity is not established. There is no correlation between fixed point metrics and standard faithfulness measures, such as deletion and insertion curves, or with human preference judgments, or with downstream task success. Without such links, the reader cannot infer whether fixed point behavior is predictive of explanation quality. The framework risks becoming a self referential audit of model explainer dynamics rather than a measure of usefulness.

**8.**
Practical aspects are under reported. The paper does not provide iteration count distributions, runtime, convergence failure rates, or sensitivity to seeds. For large models and SAEs, the cost of recursion may be high. Without cost and reliability analysis, it is difficult to recommend the method as a routine diagnostic.

**Questions:**

**Q1.**
How do the authors distinguish meaningful fixed-point convergence from trivial or degenerate cases, such as constant explainers that immediately satisfy the fixed-point condition but provide no model-specific insight? Could an additional measure of information content or explanatory sufficiency help rule out such vacuous convergence?


**Q2.**
Many explainers, including LIME and SHAP, are inherently stochastic or adaptive. How does the proposed theory extend to such cases? Could the authors analyze convergence “in expectation” across random seeds or provide empirical evidence that approximate probabilistic fixed points correlate with explanation quality?


**Q3.**
Since the recursion requires reinjecting explainer outputs through a support or masking function, how sensitive are the observed dynamics to this design choice? Would an ablation across different support mappings clarify whether convergence behavior arises from the explainer itself or from the auxiliary reinjection mechanism?


**Q4.**
Can the authors verify that convergence is not a byproduct of information collapse? For instance, could they track feature-level entropy, attribution sparsity, or retained variance across iterations to confirm that convergence reflects stability in explanatory content rather than progressive degradation?


**Q5.**
In the prototype experiments, could cycles among prototypes represent meaningful conceptual neighborhoods rather than instability? Would it be possible to evaluate whether cyclic prototype transitions align with class structure or human-interpretable clusters instead of treating them as pathological?


**Q6.**
For the SAE analysis on LLMs, can the authors examine whether recursive degradation of token predictions affects the semantic coherence of identified features? Beyond internal token preservation, would a human or qualitative evaluation help determine if the eventual P-fixed points correspond to interpretable or meaningful concepts?


**Q7.**
How do fixed-point metrics relate to established measures of faithfulness and usefulness? Could the authors report correlations with standard metrics such as deletion/insertion curves, sensitivity tests, or human judgment scores to substantiate external validity of the proposed diagnostic?


**Q8.**
What is the computational cost and reliability profile of recursive evaluation? Providing distributions of iteration counts, runtime, and convergence rates, as well as sensitivity to random seeds, would help clarify the feasibility of applying FPE to large-scale explainers or LLM settings.

---

> ### Author Response · Authors · 2025-11-22
>
> We thank the reviewer for the time they spent on articulating their concerns. We kindly ask them, before reading the rebuttal - which addresses all the points they raised - to read the message to all the reviewers and the area chairs.
>
> **W1. The core premise equates recursive stability with explanatory validity. This is conceptually fragile.**
>
> - This statement is incorrect. We respectfully ask the reviewer to read the general answer we gave to all the reviewers and the area chair. We report the salient part here.
>
> - The fixed-point and P-fixed-point explanations are presented as a framework for verifying whether the interaction between a model and its explainer is stable, faithful, and robust. By themselves, they do not guarantee these properties if the model or the explainer does not already satisfy them. Fixed-point and P-fixed points are not techniques, but a framework for explainability, as we show it applies to heterogeneous explainers and models.
>
> **W1. (continuation) The paper acknowledges faithfulness informally, but never rules out such degenerate fixed points nor provides a practical test that separates meaningful convergence from vacuous convergence.**
>
> - That is also not true. We introduce P-fixed points for this reason and test all our techniques on the premise of validating whether the model preserves the decision of the model.
>
>
> **W2. The theoretical guarantees rest on determinism and monotonicity. One could study convergence in expectation or concentration around a mean fixed point, but this is not developed.**
>
> - We would like to clarify that most of the space of the manuscript is already dedicated to introduce a novel framework for explainability, give convergence guarantees for a large class of explainers, connected it to the rich XAI literature, and report results on 3 different methods and several variations of them (SHAP, LIME, and LRP for saliency-based, self-explainable architectures for prototype-based, SAE for LLMs). While the idea the reviewer suggests makes sense, that is beyond the space of an 8-page-long paper and we strongly believe that it should be considered not as a critique, but something that we could add as future and standalone works.
>
> **W3. The recursion requires that the explainer output can be reinjected as a valid input. This domain alignment rarely holds in practice.**
>
> - We disagree with this point. Any explanation that is an endomorphism, i.e., can be re-stated as an input, can be fed to the model and explainer. This includes any heatmap-based method, NLP techniques such as Anchors, SAE, prototype-based explainers, and *any LLM-based method that gives answers in textual form*.
>
> **W4. Recursive application can collapse information and force convergence for the wrong reason. If each step removes or attenuates features, the process can converge to empty or near empty representations.**
>
> - We agree with the reviewer. However, this behavior is not a flaw of the framework, it is precisely the diagnostic purpose of our method: showing how attenuant features, which the explainer would disregard as not important, change the decision of the classifier and do not preserve the P-fixed point convergence. If the recursive process converges to empty representations, this reveals that the model-explainer interplay is not attributing relevance in a meaningful or stable way. Furthermore, in practice, our examples rarely converge to empty features, and we show that under several different support functions.

---

> > ### Author Response · Authors · 2025-11-22
> >
> > **[Continuation of the previous response]**
> >
> > **W5. The prototype analysis treats cycles as a pathology, yet cycles can reflect meaningful conceptual neighborhoods.**
> >
> > - We believe the reviewer misunderstood our claims. We do not treat cycles as pathology. In fact, Theorem 3 proves that the recursion will eventually converge to a cycle for any deterministic prototype-based model. Furthermore, we agree that cycles may reflect meaningful neighborhoods.
> > What our experiments expose is that the popular prototype-based models evaluated do not satisfy self-consistecy as defined in Definition 3.1 (i.e., that a prototype should be its own closest prototype when it is classified by the model), as we illustrate in Figure 5 and experimentally report in Table 4.
> >
> > **W6. For LLM SAEs the property P is defined as preservation of token level distributions under residual patching.**
> >
> > Thanks for your insightful comment. For SAEs the property is correctness. This is not model internal, and means that any fixed point explanation will carry the information that directly results in the model answering correctly.
> >
> > **W7. External validity is not established. There is no correlation between fixed point metrics and standard faithfulness measures.**
> >
> > - We kindly ask the reviewer to revise Section 2.1.3,  “Faithful, Stable, and Progressive Explanations”, where we establish a mathematical connection between our notions and faithfulness as introduced in [1].
> >
> > [1] D. Alvarez Melis and T. Jaakkola. Towards robust interpretability with self-explaining neural
> > networks. Neural Information Processing Systems, volume 31, 2018.
> >
> > **W8. Practical aspects are under reported. The paper does not provide iteration count distributions, runtime, convergence failure rates, or sensitivity to seeds.**
> >
> > - We respectfully disagree: we report convergence/iteration statistics for the three explanation types evaluated (see Tables 1, 2, 4 and 5). Furthermore, for each explanation type, we evaluated our framework for several explanation methods, model architectures, classification tasks and even support functions, showing consistent results across them.
> >
> > - We respectfully believe that this breadth of setups already acts as a sensitivity analysis and that it demonstrates the robustness evaluation. We also note that we provided the code to replicate all the experiments in the paper, ensuring that runtime behaviour and other particular execution metrics can be inspected and extended by the community.
> >
> > -----
> >
> > **[QUESTIONS]**
> >
> > **Q1. How do the authors distinguish meaningful fixed-point convergence from trivial or degenerate cases, such as constant explainers that immediately satisfy the fixed-point condition but provide no model-specific insight? Could an additional measure of information content or explanatory sufficiency help rule out such vacuous convergence?**
> >
> > - One of the main contributions of our paper is the notion of P-fixed point, which we discuss in great detail in the paper and covers all these concerns.
> >
> > **Q2. Many explainers, including LIME and SHAP, are inherently stochastic or adaptive. How does the proposed theory extend to such cases?**
> >
> > - Stochastic explainers (and models) admit fixed points: one can just consider each branch of the computation (defined by the probability space in which the explainer acts), and derive fixed points for probabilsitic explainers. That comes with the assumption that the explainer is monotonic: as we discussed before, relaxing that constraint would make the problem of guranteeing a fixed point an open-problem in mathematics - there are several works on AutoEncoders that discuss the convergence condition for fixed classes of neural networks that implement them; more recently, researchers are also studying Variational AutoEncoders to encompass stochasticity [1].
> >
> > [1] Sobihan Surendran et al., Theoretical Convergence Guarantees for Variational Autoencoders
> >
> > **Q3. Since the recursion requires reinjecting explainer outputs through a support or masking function, how sensitive are the observed dynamics to this design choice?**
> >
> > - We test that and report a large number of experiments for different choices of support functions in Appendix A, as we also discuss that in the main paper. Given the main contribution already present in the paper - where we introduced a novel framework for explainability, gave convergence guarantees for a large class of explainers, connected it to the rich XAI literature, and reported results on 3 different methods and several variations of them (SHAP, LIME, and LRP for saliency-based, self-explainable architectures for prototype-based, SAE for LLMs) - we preferred to move that to the Appendix but we are happy to bring it back to the main paper.

---

> > > ### Author Response · Authors · 2025-11-22
> > >
> > > **[Continuation of the previous response]**
> > >
> > > **Q4. Can the authors verify that convergence is not a byproduct of information collapse? For instance, could they track feature-level entropy, attribution sparsity, or retained variance across iterations to confirm that convergence reflects stability in explanatory content rather than progressive degradation?**
> > >
> > > - We would like to stress that this is precisely what P-fixed point explanations check, as we describe in Section 2. By testing whether a model maintains the correct classification (any other property can be added and tested, like robustness, as we discuss in the paper), we can identify precisely when a fixed point is due to information collapse.
> > >
> > > **Q5. In the prototype experiments, could cycles among prototypes represent meaningful conceptual neighborhoods rather than instability?**
> > >
> > > - Please see our response to W5.
> > >
> > > **Q6. For the SAE analysis on LLMs, can the authors examine whether recursive degradation of token predictions affects the semantic coherence of identified features?**
> > >
> > > - Thanks for the insightful question. We clarify that our method does not change the SAE in any way, and so the features retain their initial meaning (the features are coherent provided for an appropriately trained SAE [1]).
> > >
> > > [1] https://transformer-circuits.pub/2024/scaling-monosemanticity/index.html
> > >
> > >
> > > **Q7. How do fixed-point metrics relate to established measures of faithfulness and usefulness? As addressed in the weaknesses, Section 2.1.3,  “Faithful, Stable, and Progressive Explanations” explicitly discusses how fixed-point explanations and P-fixed point explanations guarantee faithfulness of the resulting explanation.**
> > >
> > > - The property (P) in the P-FPEs for SAEs is correctness. The recursive process enables us to isolate the features that directly contribute to the model’s response. In particular, the recursive process discards all activations that don’t come directly out of our SAE (and are hence directly tied to the features).
> > >
> > > - Therefore, our P-FPE is an explanation in terms of SAE feature that includes all the features the LLM might use. This is the definition of faithfulness considered in the paper.
> > >
> > > **Q8. What is the computational cost and reliability profile of recursive evaluation?**
> > >
> > > - Since our framework iteratively computes explanations, its computational cost can be straightforwardly measured based on the computational cost of the original explainer being iterated, which can vary from explainer to explainer. To complement this result, please see our response to W8.

---

> > > > ### Comment · Reviewer_uFZV · 2025-11-27
> > > >
> > > > Dear Authors,
> > > >
> > > > I appreciate the effort to address all comments. Please incorporate these new findings into the manuscript.
> > > >
> > > > At some points the claim that "**most** of the points raised by the reviewers are already addressed in the paper" appears overstated, especially given that no additional experiments were conducted during the rebuttal phase. I believe that running a few targeted experiments in this phase would help facilitate the overall process and make the response more convincing.
> > > >
> > > > Best regards,
> > > > Reviewer uFZV

---

> > > > > ### Author Response · Authors · 2025-11-28
> > > > >
> > > > > Thank you for your follow-up.
> > > > >
> > > > > In our response, we addressed all 8 weaknesses and questions raised. Since we believe most of the original concerns were conceptual in nature rather than experimental, we attempted to resolve them by clarifying the underlying concerns in our rebuttal. We would appreciate it if you could please indicate which specific experimental concerns remain unaddressed (in the paper or in the rebuttal), to ensure that our paper aligns with your expectations.

---

### Official Review · Reviewer_QTvA · 2025-11-05

**Soundness:** 3
**Presentation:** 2
**Contribution:** 2
**Rating:** 2
**Confidence:** 4

**Summary:**

This paper introduces the concept of a "fixed point explanation." The idea is to apply the classic mathematical concept of fixed points to explanation algorithms.

To arrive at the fixed point, the explanation algorithm is iteratively applied multiple times.

A high-level motivation for fixed points explanations is given: They are linked to the concept that humans, when provided with an explanation, will continue to ask further questions, that is, they will demand explanations for an explanation (this is how I understand it).

The paper presents a mathematical exposition of the properties of fixed-point explanations, including a discussion of the desirable mathematical properties of fixed points (stability, faithfulness) and the existence of fixed points.

The paper also applies the concept of fixed-point explanations to different classes of explanation algorithms (saliency maps on CIFAR-10, SAEs for language models).

Qualitative examples throughout the paper illustrate the iterative process of computing fixed-point explanations.

**Strengths:**

The idea to apply fixed-point analysis to explanation algorithms is, to the best of my knowledge, novel, and I find this idea interesting. The idea is also somewhat innovative and could be of potential interest to the community.

The mathematical exposition in the paper is sound. While most of the analysis appears to be relatively straightforward, given that the concept of fixed points is well-studied in mathematics, the overall mathematical exposition is coherent.

The concept of fixed-point analysis is applied to various explanation algorithms, including saliency maps for images and sparse autoencoders for LLMs.

**Weaknesses:**

The main weakness of the paper is that it does not contain a single empirical example where the benefits of the proposed fixed-point approach for explainability are apparent. In Figure 2, the fixed-point explanation looks just like the original explanation. In Figure 3, we can see that dynamics of iterating, but what is the point of obtaining the final image and heatmap, especially since the final image is different from the original image, and also the class label is now different? What is the real-world meaning of this explanation? In Figure 4, the result of the iteration is a strange artifact. In Figure 7, again, what are exactly are we supposed to see from these qualitative examples?

There are further limitations of the empirical evaluations; for example, there is no evaluation of standard metrics like ROAR (or any other standard metric) that compares the fixed-point explanations to other explanations (Table 1 and Table 2 evaluate the explanations according to quantitative metrics, but I am missing commonly used metric to gauge the quality of the explanations). However, the main limitation of this paper is that despite the nice mathematical exposition, there is not a single qualitative example that clearly demonstrates what the real-world benefits of the proposed approach are in an application that the community cares about. For this paper to be accepted at a conference like ICLR, multiple such examples would be required.

I also remain skeptical about the overall usefulness of the concept of fixed points when applied to explanations. Fixed points are a concept for functions that map from X to X. Most explanation algorithms, however, perform dimensionality reduction: They take the function and the input and map them to a simpler, lower-dimensional structure. The paper is aware of this complication and attempts to overcome it with a support function that maps the lower-dimensional explanation back to the input space so that the concept of fixed points can be applied. The choice of support function seems like a very critical step that deserves even more discussion than what is currently in the paper - whether the concept of fixed points is at all applicable to many explanation algorithms depends on whether we can find convincing support functions.

**Questions:**

**Question 1:** I understand that you discuss the existence of fixed points, but what about convergence? Do you have results that the iterative algorithm is guaranteed to converge to fixed points for the different setups? Can we say anything about the required number of iterations?

**Question 2:** Can you describe in intuitive terms the advantages of providing a fixed-point explanation to an end user or model developer as opposed to providing the original explanation?

**Question 3:** How does the choice of support function impact the interpretation of fixed-point explanations?

**Justification for final score:** This paper introduces the concept of fixed points to the analysis of explanation algorithms. After reading the paper, I am intrigued by this idea, but remain far from convinced that it will be fruitful. To be accepted at a venue like ICLR, this paper needs to go beyond proposing an interesting idea; it must make the case that the idea is, in fact, useful. Since the paper does not currently make this case, I recommend rejecting it at this point.

---

> ### Author Response · Authors · 2025-11-22
>
> We thank the reviewer for the time they spent on articulating their concerns. We kindly ask them, before reading the rebuttal, which addresses all the points they raised, to read the message to all the reviewers and the area chairs.
>
> **W1 - Benefits and goals of fixed-point explanations, and qualitative examples.**
> *[The main weakness of the paper is that it does not contain a single empirical example where the benefits of the proposed fixed-point approach for explainability are apparent. In Figure 2, the fixed-point explanation looks just like the original explanation. In Figure 3, we can see that dynamics of iterating, but what is the point of obtaining the final image and heatmap, especially since the final image is different from the original image, and also the class label is now different? What is the real-world meaning of this explanation? In Figure 4, the result of the iteration is a strange artifact. In Figure 7, again, what are exactly are we supposed to see from these qualitative examples?]*
>
> We thank the reviewer for this comment. We believe the concern stems from a misunderstanding of the role of fixed-point explanations in our framework. Our goal is not to propose a novel, higher-quality explanation algorithm. Instead, as we state in the abstract, our framework aims to “assess, through recursive applications, the stability of the interplay between a model and its explainer”. In other words, fixed-point explanations serve as a diagnostic tool, not as a replacement for existing explanation methods.
>
> That said, our qualitative images aim to illustrate our framework and several scenarios regarding how it captures the model-explainer interplay:
>
> Figure 2 illustrates the recursion process and the resulting fixed-point for a particular image.
>
> Figure 3 highlights a failure of class preservation: the explainer removes features that the model relies on, causing the model to change its prediction. This instability is precisely the type of behaviour our framework is designed to detect.
>
> Figure 4 aims to illustrate how recursion allows controlling explanation depth to detect steps at which the model becomes inconsistent, an insight that standard one-shot explanations cannot provide.
>
> Figure 7 contains, for each class, illustrative examples in which the original classification until convergence, showing the fixed-point features that the model thus considers minimal and sufficient for classification.
>
> Thus, these qualitative examples aim to illustrate how our framework operates and how the theoretical definitions manifest in practice, and show the behaviours that fixed-point analysis is designed to uncover: identifying inconsistencies, detecting fragile or unstable explanations, and isolating the minimal feature subsets the model ultimately relies on. We will make sure to clarify this in the manuscript.
>
> Finally, while we stress that these images are mostly illustrative, we refer the reader to our reply to W3 below, where we discuss the practical applications and actionable insights of our framework.
>
>
> **W2 - Explanation quality and comparisons.**
> *[There are further limitations of the empirical evaluations; for example, there is no evaluation of standard metrics like ROAR (or any other standard metric) that compares the fixed-point explanations to other explanations (Table 1 and Table 2 evaluate the explanations according to quantitative metrics, but I am missing commonly used metric to gauge the quality of the explanations).]*
>
> We thank the reviewer for raising this point. As stated in our earlier responses, the goal of our framework is not to introduce a new explanation method to be benchmarked against existing ones, but to provide a diagnostic tool that evaluates the interplay between a model and its explainer under recursion.
>
> Our empirical evaluation instead focuses on the properties that fixed-point analysis is specifically designed to probe: convergence, self-consistency, class preservation, and stability. Thus, our evaluations were made to validate the theoretical contributions of the work and demonstrate how fixed-point analysis can reveal inconsistencies, fragilities, or minimal sufficient feature sets, as we describe in the manuscript.
>
> That said, integrating ROAR-like metrics in future work could complement our diagnostic framework. Still, they do not evaluate the core purpose of fixed-point explanations and would not meaningfully reflect the benefits our method provides.

---

> > ### Author Response · Authors · 2025-11-22
> >
> > **[Continuation of the previous response]**
> >
> > **W3 - Practical applications.**
> > *[However, the main limitation of this paper is that despite the nice mathematical exposition, there is not a single qualitative example that clearly demonstrates what the real-world benefits of the proposed approach are in an application that the community cares about. For this paper to be accepted at a conference like ICLR, multiple such examples would be required.]*
> >
> > Our work demonstrates the practical applicability of our framework by applying our method to a wide variety of explainers, through which we have already uncovered specific inconsistencies, drawing practical and actionable findings that expose weaknesses in widely used techniques and open up several research directions. These findings are not speculative but real conclusions and thus direct demonstrations of how our approach can be used in practice to probe the robustness and consistency of existing interpretability methods, which is still a critical need in the field. This also demonstrates the meaningfulness of the recursive process: it is precisely this process that enables us to systematically probe stability, faithfulness, and consistency, as per [1], which are often assumed but rarely verified in XAI.
> >
> >
> > **W4 - Support functions.**
> > *[I also remain skeptical about the overall usefulness of the concept of fixed points when applied to explanations. Fixed points are a concept for functions that map from X to X. Most explanation algorithms, however, perform dimensionality reduction: They take the function and the input and map them to a simpler, lower-dimensional structure. The paper is aware of this complication and attempts to overcome it with a support function that maps the lower-dimensional explanation back to the input space so that the concept of fixed points can be applied. The choice of support function seems like a very critical step that deserves even more discussion than what is currently in the paper - whether the concept of fixed points is at all applicable to many explanation algorithms depends on whether we can find convincing support functions.]*
> >
> > First, we would like to express that $\epsilon$ is naturally an endomorphism for prototype-based explainers and SAEs. However, we acknowledge the practical relevance of the support function in the case of feature-based explainers. For this reason, in line with most of the standard perturbation- and occlusion-based saliency methods (which are widely used to probe feature importance in image classifiers), we adopt masking as support functions for feature-based classifiers [1,2,3,4,5,6]. These methods commonly remove information by blurring or replacing selected regions with a constant baseline (e.g., zero or grey), a choice that aligns with our default support function (zero-out). We will make this justification explicit in the paper and include the relevant references that demonstrate the prevalence of masking in the XAI literature.
> >
> > Furthermore, Appendix A includes an explicit ablation study across multiple support functions for feature-based explainers, described in that appendix and illustrated in Figure 8.  We also justify these choices on principled grounds: they strike a deliberate balance between effective feature suppression and maintaining visual coherence, which serves to reinforce the robustness of our conclusions.
> > Our results show consistent behaviour across all tested supports regarding convergence and model consistency, indicating that our conclusions are robust to the choice of $s$.
> >
> > Finally, we would like to emphasise that our goal in Section 2 was to introduce a general and unifying theoretical framework applicable to a wide range of explainer types – a strength highlighted by all the reviewers. A key aspect of this generality is that the theory does not depend on the specific choice of the support function. This is why our formal definitions and propositions are stated independently of any particular instantiation of $s$.

---

> > > ### Author Response · Authors · 2025-11-22
> > >
> > > **[Continuation of the previous response]**
> > >
> > > ------
> > >
> > > **QUESTIONS**
> > >
> > > **Q1: I understand that you discuss the existence of fixed points, but what about convergence? Do you have results that the iterative algorithm is guaranteed to converge to fixed points for the different setups? Can we say anything about the required number of iterations?**
> > >
> > > We discuss convergence for feature-based explainers in Section 3.1 (under the monotinicity of $\epsilon \circ s$), and provide a convergence guarantee for prototype-based models in Theorem 3 (Section 3.2). We also discuss the convergence for contractive SAEs in Appendix C. Furthermore, we empirically validated convergence in our experimental results in Section 4. For instance, Table 1 reports the convergence steps for LIME, SHAP and LRP across three image classification tasks. Similarly, we report convergence statistics for prototype-based models in Table 4 (Appendix B.3).
> > >
> > >
> > > Apart from that, convergence is guaranteed by many fixed point theorems in math (Brouwer, Kakutani, or Tarski, the one we used and fits with feature-based explainers). Guarantees for non-monotonic explainers relate to guarantees for arbitrary functions, which is an open problem in math and for which it is well known that some classes of functions do not converge (indeed, they diverge, as we show for the case of non-contractive SAEs, in Appendix C).
> > >
> > > **Question 2: Can you describe in intuitive terms the advantages of providing a fixed-point explanation to an end user or model developer as opposed to providing the original explanation?**
> > >
> > > We would like to clarify that our framework does not substitute existing explainability tools, but integrates them in a dynamic framework that allows us to check whether one should really trust the explanation. For example, let us suppose that a user employs SHAP and gets an explanation for an image that is correctly classified as a cat. Then, according to our framework, the user decides to test the explanation against the model and receives a different label (e.g., dog). Would the user trust the original explanation given that the model provides a different label for it?
> > > In general, letting the model+explainer interact and converge gives us information about how the model treats the salient features of the explanations and what features are sufficient to imply the decision of a model.
> > >
> > > **Question 3: How does the choice of support function impact the interpretation of fixed-point explanations?**
> > >
> > > Please see the answer to W4.
> > >
> > >
> > > **Justification for final score:**
> > > *This paper introduces the concept of fixed points to the analysis of explanation algorithms. After reading the paper, I am intrigued by this idea, but remain far from convinced that it will be fruitful. To be accepted at a venue like ICLR, this paper needs to go beyond proposing an interesting idea; it must make the case that the idea is, in fact, useful. Since the paper does not currently make this case, I recommend rejecting it at this point.*
> > >
> > > We hope this rebuttal, alongside the message we shared among all the reviewers, will improve your judgment about our paper. Thank you again for your time.
> > >
> > > **References**
> > >
> > > [1] Interpretable Explanations of Black Boxes by Meaningful Perturbation
> > >
> > > [2] Explaining Image Classifiers by Removing Input Features Using Generative Models
> > >
> > > [3] “Why Should I Trust You?” Explaining the Predictions of Any Classifier
> > >
> > > [4] Visualizing and Understanding Convolutional Networks
> > >
> > > [5] Understanding Deep Networks via Extremal Perturbations and Smooth Masks
> > >
> > > [6] Axiomatic Attribution for Deep Networks
> > >
> > > [7] RISE: Randomized Input Sampling for Explanation of Black-box Models

---

> > > > ### Comment · Reviewer_QTvA · 2025-11-25
> > > > **Response to Authors**
> > > >
> > > > Thank you for the detailed rebuttal and responses to my questions.

---

> > > > > ### Author Response · Authors · 2025-11-26
> > > > >
> > > > > Thank you for taking the time to read our rebuttal. Given the substantial effort we put into addressing each of the concerns you raised, we would greatly appreciate any further feedback you may have regarding the clarifications we provided, or whether the rebuttal affects your assessment of the paper.

---

### Official Review · Reviewer_yLGy · 2025-11-06

**Soundness:** 2
**Presentation:** 2
**Contribution:** 2
**Rating:** 2
**Confidence:** 3

**Summary:**

The paper presents an effort to analyze the properties of explainability methods using a regressive framework, where explanations are recursively applied until convergence. Several case studies are provided to illustrate how such recursive applications can reveal properties like consistency and stability across different types of explainability methods.


However, the paper tends to be somewhat abstract in parts, and certain subsections suffer from notational inconsistencies that make them difficult to follow. Moreover, the practical utility of this work is questionable in the current landscape, where explainability has moved towards multimodal paradigms (for example, visual decisions explained through text). The study remains confined to feature-based, saliency-driven explanation frameworks.

**Strengths:**

The authors present solid arguments for analyzing explanation methods through the “why regress” principle, focusing on consistency and stability as key properties.

The proposed framework is explored across both feature-based explainers such as VGG16 and transformer-based language models, showing some versatility in application.

**Weaknesses:**

The work is limited to saliency or feature-attribution-based explanation methods, most of which are relatively dated (5–7 years old). While understanding their properties is intellectually useful, the motivation feels weak in today’s context. Modern vision-language models no longer rely on such feature-based explanations, raising questions about the current relevance of the work.

The description of the SAE’s application on transformer models lacks clarity. In particular, it is not well explained why the SAE was applied to the last-layer outputs (before tokenization).

The theoretical guarantees of convergence and other claimed properties are based on assumptions that may not hold for many practical XAI methods.

Computational complexity is not discussed at all, despite the authors mentioning recursive applications extending up to 20 or more steps in some cases.

Prototype-based explainers are not shown to satisfy consistency or stability, and the authors defer this as “future work.” This weakens the overall claim of analyzing fixed-point, regression-based explanation methods.

Figure 4 is confusing. The recursive steps alter class labels, and the final “converged” explanation appears inconsequential. The visualization raises more questions than it answers.

**Questions:**

The writing needs significant refinement. Several sections are incomprehensible and overloaded with notation that is neither introduced nor explained clearly.

The paper does not adequately justify the choice of support functions or how they are constructed, merely referring to them as a “common choice.”

The recursive procedure might violate the input data distribution on which the original models were trained, an issue that remains unaddressed.

The discussion on convergence guarantees under non-monotonic conditions (a notably strict requirement) is superficial, representing a major oversight.

Overall, the positioning of the work is unclear. The paper does not articulate what key takeaways modern XAI methods should derive from this study to enhance their practical applicability.

---

> ### Author Response · Authors · 2025-11-22
>
> We thank the reviewer for the time they spent on articulating their concerns. We kindly ask them, before reading the rebuttal, which addresses all the points they raised, to read the message to all the reviewers and the area chairs.
>
> **The work is limited to saliency or feature-attribution-based explanation methods, most of which are relatively dated (5–7 years old). While understanding their properties is intellectually useful, the motivation feels weak in today’s context.**
>
> - This statement is objectively incorrect. A key strength of our work, as highlighted by multiple reviewers, is precisely that our framework is not limited to feature-attribution methods. Section 3 explicitly develops the recursive formulation for three distinct families of explainers:  feature-based explainers (e.g., LIME, SHAP, LRP), prototype-based explainers and mechanistic explainers such as Sparse Autoencoders (SAEs). In Section 4, we provides experimental validation across particular methods for the three families, including methods such as LIME, SHAP and LRP (three of the most popular feature-attribution methods), two self-explainable prototype-based models, and SAEs, demonstrating that our fixed-point framework applies beyond traditional saliency maps.
>
> - Modern vision-language models no longer rely on such feature-based explanations, raising questions about the current relevance of the work.
>
> - Based on the arguments we provide in what follows, we strongly believe that this is an overstatement and an LLM-centric view of explainability, which encompasses many more fields than just LLMs. Regarding saliency-based methods, LIME is cited around 6.000 times this year, and the works that mention or use it for LLMs are only a fraction. The same is true for SHAP, cited 14.000 times this year.
>
> - A similar case can be made for SAEs, which are, indeed, a very active area of research in the LLM ecosystem and thus cover up-to-date explainability techniques.
> Furthermore, our framework seamlessly extends to vision-Transformers: one can consider the input to the model to be the image and the caption, and the textual output as the target of the explainability method. Indeed, SAEs have been applied to vision transformers [1]
>
>
> [1] Vladimir Zaigrajew, Hubert Baniecki, & Przemyslaw Biecek (2025). Interpreting CLIP with Hierarchical Sparse Autoencoders. In the Forty-second International Conference on Machine Learning.
>
>
> **The description of the SAE’s application on transformer models lacks clarity. In particular, it is not well explained why the SAE was applied to the last-layer outputs (before tokenization).**
>
> - We want to clarify that we do not apply it to the last layer. **Could the reviewer specify where they read this?**
>
> - Indeed, we specify that we apply it to a deep layer, which, in particular, is layer 50 (out of 80). The choice is justified by prior work [1] showing that the very last layers tend to deviate significantly from the rest of the model (presumably due to operating directly on the logits, as in adjusting probabilities instead of computing representations of the input). We will make sure to include this detail in the manuscript.
>
> [1] Davide Ghilardi, Federico Belotti, Marco Molinari, and Jaehyuk Lim. 2024. Accelerating Sparse Autoencoder Training via Layer-Wise Transfer Learning in Large Language Models. In Proceedings of the 7th BlackboxNLP Workshop: Analyzing and Interpreting Neural Networks for NLP, pages 530–550, Miami, Florida, US. Association for Computational Linguistics.
>
> **The theoretical guarantees of convergence and other claimed properties are based on assumptions that may not hold for many practical XAI methods.**
>
> - Our theoretical guarantees hold for several widely used classes of explainers. In particular, our convergence and fixed-point results apply to three distinct explainer families developed in Section 3—feature-based explainers, prototype-based explainers, and mechanistic explainers such as Sparse Autoencoders—and Section 4 provides empirical evidence across multiple algorithms within each family. These guarantees rely on assumptions (e.g., determinism and monotonicity under the support function) that are satisfied by many practical XAI methods and that enable us to establish a unified and mathematically grounded theory of recursive explanations.
>
> - Naturally, some explainers – such as modern LLMs and LVLMs and other non-monotonic or stochastic systems – introduce additional complexities that require dedicated analysis and theoretical tools. To the best of our knowledge, this work is the first of its kind. Thus, we believe that establishing these foundations is a valuable and necessary step toward extending the theory to more complex settings. We explicitly acknowledge this and view it as an exciting direction for future work, and consider that the scope of this framework is already reasonable.

---

> > ### Author Response · Authors · 2025-11-22
> >
> > **[Continuation from the previous response]**
> >
> > **Computational complexity is not discussed at all, despite the authors mentioning recursive applications extending up to 20 or more steps in some cases.**
> >
> > - We are happy to add that part to the revised version of the paper. We will use the Master theorem, or a similar approach, to provide the computational complexity of our framework. As a note, recursively applying a SAE (or another explainer) is a linear function of how expensive explainer itself is (which in the case of the SAE means we are able to reach fpes in less than a minute on 1 GPU even for the SAE of Llama 70b). Our method is O(n_recursions*explainer_cost).
> >
> > **Prototype-based explainers are not shown to satisfy consistency or stability, and the authors defer this as “future work.” This weakens the overall claim of analyzing fixed-point, regression-based explanation methods.**
> >
> > - We respectfully disagree with the reviewer’s interpretation. Our framework is explicitly designed to diagnose consistency and stability, not to fix them. In fact, we view the reviewer’s concern as reinforcing the relevance of our findings: the fact that prototype-based models do not satisfy stability or consistency underscores that these vulnerabilities are real and insufficiently examined in current XAI practice. In other words, the fact that the reviewer considers our finding something to be addressed reveals that our framework was capable of exposing a real, relevant limitation in those models. However, we stress again that our framework is specifically designed to surface such issues. Addressing and repairing these model-specific deficiencies would require substantial, dedicated methodological development, which would overextend the scope of this work, and which, therefore, is deferred as an important direction for future work.
> >
> > **Figure 4 is confusing. The recursive steps alter class labels, and the final “converged” explanation appears inconsequential. The visualization raises more questions than it answers.**
> >
> > - Figure 4 shows a case where the interplay between a model and its explanation is brittle, yet converges to an answer that the model eventually classifies as the right class. We believe this image provides insights and poses questions on the stability of many explainers (in this case, SHAP), something that other explainability frameworks (including using SHAP as a one-shot method) lack.
> >
> >
> > **The writing needs significant refinement. Several sections are incomprehensible and overloaded with notation that is neither introduced nor explained clearly.**
> >
> > - We appreciate the reviewer’s concern regarding clarity. We will ensure that all notation is introduced, defined, and explained at the point of first use.
> >
> >
> > **The paper does not adequately justify the choice of support functions or how they are constructed, merely referring to them as a “common choice.”**
> >
> > - In line with most of the standard perturbation- and occlusion-based saliency methods, which are widely used to probe feature importance in image classifiers, we adopt masking as support functions for feature-based classifiers [1,2,3,4,5,6]. These methods commonly remove information by blurring or replacing selected regions with a constant baseline (e.g., zero or grey), a choice that aligns with our default support function (zero-out). Furthermore, in Appendix A, we also evaluate several alternative support functions (Gaussian blurring, exponential decay and class-wise average filling) and describe each, including qualitative illustrations in Figure 8.
> >
> > - We also justify these choices: they strike a balance between effectively suppressing irrelevant features and maintaining visual coherence, which serves to reinforce the robustness of our conclusions.  We will make these justifications explicit in the paper and include the relevant references that demonstrate the prevalence of masking in the XAI literature.
> >
> >
> > **The recursive procedure might violate the input data distribution on which the original models were trained, an issue that remains unaddressed.**
> >
> > - This issue is discussed in Appendix A, where we acknowledge the difficulty of designing support functions, particularly for feature-based methods: they must effectively suppress discarded features without excessively distorting the input or reintroducing discarded features. To test the sensitivity of our framework to this choice, we conducted additional experiments exploring several support functions for feature-based explanations on CIFAR-10 (the methods are listed and described in Appendix A). While fitting our full discussion in the main body might be unfeasible due to the page limit, we will emphasise this matter in the main paper to address this concern.

---

> > > ### Author Response · Authors · 2025-11-22
> > >
> > > **[Continuation from the previous response]**
> > >
> > > **The discussion on convergence guarantees under non-monotonic conditions (a notably strict requirement) is superficial, representing a major oversight.**
> > >
> > > - We thank the reviewer for highlighting this point. We agree that non-monotonic explainers present an interesting research avenue, as we explicitly acknowledge and position as future work.
> > >
> > > - However, our goal in this paper is to establish the first general theoretical framework for recursive explainability and fixed-point analysis. To do so rigorously, we adopt assumptions – e.g., monotonicity – that allow us to prove existence and convergence results and to derive meaningful properties. These assumptions are not meant to describe all explainers, but to provide the foundational layer on top of which more complex cases can be studied. Nonetheless, our framework already covers a broad and practically relevant class of explainers (feature-based, prototype-based, and SAEs), as shown in Section 3 and validated experimentally in Section 4.
> > >
> > > - Thus, extending convergence guarantees to other explainers such as LLMs requires a mathematical formulation that goes beyond the scope of the paper, as one needs to introduce, for the case of LLMs and free-text form, non-monotonic reasoning and some sort of stability like Lyapunov functions. Therefore, a more rigorous treatment of these non-monotonic cases demands theory and tools that fall outside the scope of this paper, which is why our discussion can only be high-level to motivate this interesting future research path.
> > >
> > > **Overall, the positioning of the work is unclear. The paper does not articulate what key takeaways modern XAI methods should derive from this study to enhance their practical applicability.**
> > >
> > > - We demonstrated the practical applicability of our framework by applying our method to a wide variety of explainers, through which we have already uncovered specific inconsistencies, drawing practical and actionable findings that expose weaknesses in widely used techniques and open up several research directions. These findings are not speculative but real conclusions and thus direct demonstrations of how our approach can be used in practice to probe the robustness and consistency of existing interpretability methods, which is still a critical need in the field. This also demonstrates the meaningfulness of the recursive process: it is precisely this process that enables us to systematically probe stability and consistency, as per [7], which are often assumed but rarely verified in XAI.
> > >
> > > **References**
> > >
> > > [1] Interpretable Explanations of Black Boxes by Meaningful Perturbation
> > > [2] Explaining Image Classifiers by Removing Input Features Using Generative Models
> > > [2] “Why Should I Trust You?” Explaining the Predictions of Any Classifier
> > > [3] Visualizing and Understanding Convolutional Networks
> > > [4] Understanding Deep Networks via Extremal Perturbations and Smooth Masks
> > > [5] Axiomatic Attribution for Deep Networks
> > > [6] RISE: Randomized Input Sampling for Explanation of Black-box Models
> > > [7] Towards robust interpretability with self-explaining neural networks. NeurIPS'18

---

### Official Review · Reviewer_6LhJ · 2025-11-08

**Soundness:** 2
**Presentation:** 3
**Contribution:** 3
**Rating:** 2
**Confidence:** 3

**Summary:**

The paper's main idea is to test how stable an explanation is by recursively explaining the explanation. This is inspired by the "why regress?" principle. Based on my understanding, this means:
1. You get an explanation for an input (e.g., a heatmap of important pixels).
2. You use that explanation as the new input for the explainer.
3. You repeat this process until the explanation stops changing.

This final, unchanging explanation is what the authors call a "fixed point". They argue this fixed point should be a minimal and faithful explanation. They also check if a key property (like the model's prediction staying correct) holds true during every step; if it does, it's a "P-fixed point". The authors use this method as a "sanity check" to find hidden instabilities in different XAI tools, including LIME, SHAP, and even methods for LLMs like SAEs.

**Strengths:**

1. The concept of a "fixed point explanation," based on recursively applying an explainer to its own output (the "why regress" principle), is an interesting new way to think about and evaluate the stability of XAI methods.
2. The paper's ambition in applying this single framework across three very different and timely classes of explainers (feature-based, prototype-based, and mechanistic) demonstrates the potential generality of the fixed-point concept.

**Weaknesses:**

1. **Unclear Core Methodology and Role of the Support Function**:  I found the paper's central definition difficult to follow. The main methodology in Section 2 defines the recursive step as $x_k = \epsilon(x_{k-1}; f)$, which implies the explainer $\epsilon$ outputs a new object in the same format as the original input $x$. However, for feature-based explainers like LIME or SHAP (Section 3.1), the explainer's output isn't a new image, but rather a set of feature importance or a heatmap (which the paper calls $Z$). To make this work, the paper briefly introduces a "support function" $s$ to map this explanation ($Z$) back to an input ($X$). This support function seems critical to the whole process, but it's absent from the main definitions (2.1, 2.2, 2.3) and only discussed in detail in the Appendix. This makes it very confusing to understand what is actually being iterated. It seems the choice of $s$ (e.g., zeroing out, blurring) is just as important as the explainer $\epsilon$, but it's not treated as a core part of the method. Clarifying how $s$ interacts with $\epsilon$, and whether all experiments use the same $s$, would greatly improve the paper’s clarity.

2. **Conceptual gap in what the "explanation" is**: The paper claims the final fixed point $x*$ is a minimal, stable, and faithful explanation. However, looking at the examples (e.g., Figure 2 or Figure 7), the fixed point $x^*$ doesn't look like an "explanation" in the sense of why the model made its decision. Instead, it looks like a minimal input that can still trigger the same classification (e.g., the glowing outlines of the digits in Figure 7). It's not clear how this is a new type of explanation, as opposed to a different way of finding a "sufficient input subset," which is a known concept. The paper's "why regress" justification was hard to connect to the actual process, which seems to be more about iterative feature removal than iteratively deepening an answer.

3. **Mismatch Between “P-Fixed Point” Definition and Experimental Outcomes**: A major contribution seems to be the "P-fixed point explanation" (Definition 2.4), which acts as a "certificate" that a property $P$ (like correct classification) holds at every step. This is presented as a desirable outcome. However, the empirical results often show this property failing.

    (a) In Fig. 4, the classification shifts from Sneaker $\rightarrow$ Sandal $\rightarrow$ Sneaker during recursion.

    (b) In Fig. 3, it changes from Airplane $\rightarrow$ Bird.

    If the main goal of the P-fixed point concept is to maintain such properties, then its failure undermines its practical utility. While the authors argue these failures are “insights,” it remains unclear what actionable interpretability insight is gained when the defining property of the explanation does not hold. Clarifying whether instability is expected (and why) would make the framework’s intent clearer.

4. **Questionable conclusions from SAE experiments:** The paper applies its recursive method to Sparse Autoencoders (SAEs) and finds that the LLM's correctness drops to zero. The paper then claims this poses serious issues with the SAEs themselves and their ability to explain the model. I am not sure this conclusion is justified. The FPE method involves taking the SAE's reconstruction, $z_k$, and feeding it back into the SAE to get $z_{k+1}$. This recursive process seems to push the hidden activations far away from their original state (as shown by the very low Jaccard similarity and high KL divergence in Table 2). It's possible this failure isn't an "explanatory weakness" of the SAE, but simply an artifact of using the SAE in a way it was never trained for (i.e., on inputs that don't come from the LLM's actual hidden states).

**Questions:**

Please refer to weaknesses.

---

> ### Author Response · Authors · 2025-11-22
>
> We thank the reviewer for the time they spent on articulating their concerns. We kindly ask them, before reading the rebuttal, which addresses all the points they raised, to read the message to all the reviewers and the area chairs.
>
>
> **W1 -  Unclear Core Methodology and Role of the Support Function:**
>
> - First, we would like to emphasise that our goal in Section 2 was to introduce a general and unifying theoretical framework applicable to a wide range of explainer types – a strength highlighted by all the reviewers. A key aspect of this generality is that the theory does not depend on the specific choice of the support function. This is why our formal definitions and propositions are stated independently of any particular instantiation of $s$. In line with our design choice, and given the strict page limit, we (1) placed the deeper technical discussion of support functions in Appendix A, and (2) deferred explainer-specific choices of $s$ to the explainer-specific sections in Section 3. Nevertheless, we will include a deeper discussion on $s$ at the beginning of Section 2, which will better position its importance and role in our theory.
>
> - Finally, we also stress that, while $\epsilon$ is naturally an endomorphism for prototype-based explainers and SAEs, we acknowledge the practical relevance of the support function in the case of feature-based explainers. Because of this, our experiments include an explicit ablation across multiple support functions for feature-based explainers. These results show consistent behaviour across all tested supports regarding convergence and model consistency, indicating that our conclusions are robust to the choice of $s$.
>
>
> **W2 - Conceptual gap in what the "explanation" is:**
>
> - We believe the concern stems from a misunderstanding of the role of fixed-point explanations in our framework. Our goal is not to propose a new user-facing explanation algorithm, nor to claim that the fixed point itself is always semantically rich. Rather, as we state in the abstract, our framework aims to “assess, through recursive applications, the stability of the interplay between a model and its explainer”. In other words, fixed-point explanations serve as a diagnostic tool, not as a replacement for existing explanation methods.
>
> - Regarding “it looks like a minimal input that can still trigger the same classification [...]. It's not clear how this is a new type of explanation”. We stress that this outcome is precisely a consequence of our recursive framework: the fixed point represents the subset of features that the model itself deems sufficient and minimal after repeatedly discarding those marked as irrelevant by the explainer. In other words, the fixed point is not intended to be a human-interpretable explanation on its own, but rather the result of a principled recursive process that reveals (1) what the model ultimately considers essential for that specific prediction, and (2) the stability of the explainer-model interplay.
>
>
> **On the “why regress” connection:**
>
> - The reviewer suggests that the process “seems to be more about iterative feature removal than iteratively deepening an answer.” However, the “iterative feature removal” is the operationalisation of the “why regress” principle for feature-based explainers. In human reasoning, repeatedly asking “why?” removes secondary details until only the essential cause remains. In our framework, recursive explanation removes non-essential features until only the minimal explanation that the explainer and model agree upon persists.

---

> > ### Author Response · Authors · 2025-11-22
> >
> > **[Continuing from the previous response]**
> >
> > **W3 - Mismatch Between “P-Fixed Point” Definition and Experimental Outcomes:**
> >
> > - We thank the reviewer for raising this point and are happy to clarify the intent behind the notion of P-fixed points. The reviewer’s concern arises from an implicit assumption that P-fixed points are meant to always hold in practice. Instead, *our framework is explicitly designed so that violations of the property during recursion are informative outcomes – not failures of the method*. Our contribution is an explainability framework that highlights failures and brittleness of the interplay between a model and its explanation, not an explainability technique that comes with guarantees of preserving the output class.
> >
> > - In other words, P-Fixed Points” prescribe as ideal the case where a property (e.g., class preservation) is maintained at every recursive step. The recursive process is a test of whether an explainer maintains the property under repeated application, not a guarantee that it should. When a property fails (e.g., class flips in Figs. 3–4), this exposes a specific misalignment between the explainer and the model – a phenomenon that existing XAI literature rarely measures. For instance, while it is well known that modern explainers are brittle, there is no standard tool to quantify or trace how and when this happens. Our recursive framework reveals this instability step-by-step, offering a systematic sanity check on whether the model-explanation interplay remains consistent.
> >
> > - We will clarify all these matters in the paper, in order to highlight the practical and actionable insights that our framework brings to the community.
> >
> >
> > **W4 - Recursive SAE interplay with LLM.**
> >
> > - We thank the reviewer for his observation, but we’d like to point out that the SAE is trained to output to and from the LLM’s activation space. The inputs and the expected outputs during SAE training are activations in that space. Therefore, by the properties of vector spaces, a perfect SAE should output vectors in its own input space.
> >
> > - In practice, this does not happen (the seminal SAE paper theorised that superposition in LLM internal representations scales exponentially with parameter count [1], which means you’d need an astronomically large SAE to achieve perfect reconstruction).
> > In the absence of perfect SAEs, evals have been proposed to quantify the fidelity of an LLM’s reconstruction [2].
> > Our framework is informative insofar as it gives guarantees that SAE features are used by the LLM, properties of minimality and faithfulness, and certificates that hold asymptotically (such as correctness).
> >
> > - Overall, our study provides a way to enhance trust in a method (SAEs), the efficacy of which has recently been called into question [3]. In particular, in terms of minimality and faithfulness, but also in terms of certificates that users can specify based on the specific task being studied.
> >
> >
> > **References**
> >
> > [1] Problem formulation in: https://transformer-circuits.pub/2023/monosemantic-features/index.html
> >
> > [2] Aleksandar Makelov, Georg Lange, & Neel Nanda (2025). Towards Principled Evaluations of Sparse Autoencoders for Interpretability and Control. In The Thirteenth International Conference on Learning Representations.
> >
> > [3] https://www.alignmentforum.org/posts/4uXCAJNuPKtKBsi28/sae-progress-update-2-draft

---

> > > ### Comment · Reviewer_6LhJ · 2025-11-26
> > >
> > > I thank the authors for addressing my technical concerns regarding the methodology and the SAE results. I accept the premise that this is intended as a diagnostic framework. However, the manuscript itself remains difficult to read. The paper needs to be more intuitive and self-contained than it currently is. I am willing to raise my score to 4 (after the submission is updated) to acknowledge the explanation provided in the rebuttal, but I believe the paper requires significant restructuring to effectively communicate its contribution.

---

### Author Response · Authors · 2025-11-22
**Message to the reviewers, ACs, SACs, and PCs**

We thank the reviewer for their response.


However, we believe several points in the reviews reflect misunderstandings of the paper, and a number of claims are not supported by what the article actually presents, proves, or evaluates. Below we list the major misinterpretations we identified (particularly in the reviews by 6LhJ, yLGy, and QTvA). We also found it difficult to address the weaknesses and questions of reviewer uFZV, as most of the points they raise are indeed already addressed in the paper.


- The fixed-point and P-fixed-point explanations are presented as a framework for verifying whether the interaction between a model and its explainer is stable, faithful, and robust. By themselves, they do not guarantee these properties if the model or the explainer does not already satisfy them. Fixed-point and P-fixed points are not explanation techniques, but a framework for explainability, as we show it applies to wide range of explainers and models.


- We assume that an explainer is deterministic and finite to establish the theoretical foundations of our framework. On the other hand, for probabilistic explainers and models, being arbitrary functions, guaranteeing convergence to fixed points is an open mathematical problem beyond the scope of this paper and would constitute a significant advance in analysis. On the other hand, when both the model and explainer are probabilistic but finite and monotonic, each probabilistic “branch” still converges to a fixed point (a direct consequence of treating each branch as deterministic).


- We discuss the role of the support function in depth. The main motivations are introduced in the paper, and Appendix A provides a detailed explanation of their importance. We dedicate an entire section, including experiments, to several types of support functions, showing how our framework encompasses widely used techniques (from occlusion to noise-based methods).


- The experiments are not limited to feature-based models. As shown in the evaluation, they include segmenters (along with two widely used explainers in the ML literature, LIME and SHAP), prototype-based explanations (with a full dedicated section), and also experiments involving Large Language Models and Sparse Autoencoders. Some reviewers (e.g., yLGy) pointed out that saliency-based methods are not up to date with the current literature, where vision-Transformers models have multi-modal inputs: that’s an LLM-centric view of explainability, which encompasses many more fields than just LLMs. LIME is cited around 6000 times this year, and the works that mention or use it for LLMs are only a fraction. The same is true for SHAP, cited 14.000 times this year. Reviewer yLGy also adds as a weakness that our work is limited to saliency-based methods, a statement that is inconsistent with the content of the paper, which includes four full subsections (Sections 3.2, 3.3, 4.2 and 4.3) dedicated to the other two methods described above (and many more to add in the future, consider LLMs which can be treated as endomorphism as they are text-to-text generators). Since these components are central to the contribution, this comment suggests a misunderstanding or an incomplete reading of the manuscript.


For these reasons, we respectfully ask the reviewers to reconsider their evaluations and request that the Area Chairs oversee this process to ensure a fairer review and discussion phase.

---

### Meta-Review · Area_Chair_pZHR · 2025-12-08

**Summary:**

This paper introduces "fixed point explanations" - a framework that recursively applies an explainer to its own output until convergence, inspired by the philosophical "why regress?" principle. The authors formalize this concept mathematically, define convergence conditions for feature-based, prototype-based, and mechanistic (SAE) explainers, and argue that fixed points reveal stability properties of the model-explainer interplay.

**Reviewer Concerns:**

1. Unclear practical utility and usage of the explanation
The reviewers raised concerns on the actionable insight the fixed point provides.
2. Role of support functions
Reviewers noted the support function (mapping explanations back to inputs) is critical but underexplained in the main text.
3. Stochasticity and monotonicity assumptions
The theoretical guarantees require determinism and monotonicity, which many practical explainers violate. The authors acknowledge this as a limitation and future work direction.
4. Interpretation of "failures"
When P-fixed points fail (class changes during recursion), reviewers questioned whether this is informative or just expected behavior.
5. SAE experiments
Concerns that applying SAEs recursively uses them outside their training distribution.


The authors provided substantial rebuttals clarifying that this is a diagnostic framework rather than a new explanation method, and that "failures" (e.g., class changes during recursion) are precisely the diagnostic insights the framework is designed to surface.

While the paper introduces a conceptually new  framework with mathematical foundations, the reviews reveal significant concerns about practical utility and presentation clarity that were only partially resolved through rebuttal.  Specifically, the authors' rebuttal reframes their work as a diagnostic tool, which contradicts the paper's stronger claim of presenting a new class of explanation with guaranteed properties. This fundamental ambiguity about the contribution's nature is a central weakness, as the manuscript is not structured or evaluated effectively as a diagnostic. The paper would benefit from substantial revision.

**Reviewer Scores:**

6LhJ: 2->4 (explicitly stated):  some concerns are addressed.
yLGy:2->4. Made imprecise claim that work such as  "limited to saliency methods" and  "last layer". Authors addressed some concerns.
QTvA:2->2. core concern about practical utility demonstration remains.
uFZV:4->4/6.  concerns are partially addressed.
WWxa:6->6.  maintain positive.

---

### Decision · Program_Chairs · 2026-01-26

Reject